# COMPRESSED ONLINE SINKHORN

### ABSTRACT

The use of optimal transport (OT) distances, and in particular entropic-regularised OT distances, is an increasingly popular evaluation metric in many areas of machine learning and data science. Their use has largely been driven by the availability of efficient algorithms such as the Sinkhorn algorithm. One of the drawbacks of the Sinkhorn algorithm for large-scale data processing is that it is a two-phase method, where one first draws a large stream of data from the probability distributions, before applying the Sinkhorn algorithm to the discrete probability measures. More recently, there have been several works developing stochastic versions of Sinkhorn that directly handle continuous streams of data. In this work, we revisit the recently introduced *online Sinkhorn algorithm* of Mensch & Peyré (2020). Our contributions are twofold: We improve the convergence analysis for the online Sinkhorn algorithm, the new rate that we obtain is faster than the previous rate under certain parameter choices. We also present numerical results to verify the sharpness of our result. Secondly, we propose the *compressed online Sinkhorn algorithm* which combines measure compression techniques with the online Sinkhorn algorithm. We provide numerical experiments to show practical numerical gains, as well as theoretical guarantees on the efficiency of our approach.

## 1 INTRODUCTION

A fundamental problem in data processing is the computation of metrics or distances to effectively compare different objects of interest (Peyré et al., 2019). In the last decade, it has become apparent that many problems, including image processing (Ferradans et al., 2014; Ni et al., 2009), natural language processing (Xu et al., 2020) and genomics (Schiebinger et al., 2019), can be modelled using probability distributions and optimal transport (OT) or *Wasserstein distances* have become widely adopted as evaluation metrics. The use of such distances have become especially prevalent in the machine learning community thanks to the vast amount of research in computational aspects of entropic-regularised optimal transport (Cuturi, 2013; Peyré et al., 2019). The use of entropic-regularised optimal transport has been especially popular since they can be easily computed using the celebrated Sinkhorn algorithm (Cuturi, 2013), and such distances are known to have superior statistical properties, circumventing the curse of dimensionality (Genevay et al., 2019).

Given the wide-spread interest in computational optimal transport and in particular, entropic regularised distances in large data-processing applications, there have been several lines of work on extending the Sinkhorn algorithm to handle large-scale datasets. The application of the Sinkhorn algorithm typically involves first drawing samples from the distributions of interest and constructing a large pairwise distance matrix, then applying the Sinkhorn algorithm to compute the distance between the sampled empirical distributions. While there have been approaches to accelerate the second step with Nyström compression (Altschuler et al., 2019) or employing greedy approaches (Altschuler et al., 2017), in recent years, there has been an increasing interest in the development of online versions of Sinkhorn that can directly compute OT distances between continuous distributions. One of the computational challenges for computing the OT distance between continuous distributions is that the dual variables (Kantorovich potentials) are continuous functions and one needs to represent these functions in a discrete manner. Two of the main representations found in the literature include the use of reproducing kernel Hilbert spaces (Aude et al., 2016) and more recently, the online Sinkhorn algorithm (Mensch & Peyré, 2020) was

introduced where the Kantorovich potentials are represented using sparse measures and special kernel functions that exploit the particular structure of OT distances.

In this work, we revisit the online Sinkhorn algorithm of Mensch & Peyré (2020) and make two contributions. **Contribution 1.** We provide an updated theoretical convergence rate that, under certain parameter choices, is faster than the rate given in Mensch & Peyré (2020). We also numerically verify that our theoretical analysis is sharp. **Contribution 2.** We propose a compressed version of the online Sinkhorn algorithm. The computational complexity of the online Sinkhorn algorithm grows polynomially with each iteration and it is natural to combine compression techniques with the online Sinkorn algorithm. As explained in Section 3, the online Sinkhorn algorithm seeks to represent the continuous Kantorovich potentials as measures (super-position of Diracs). We apply compression with a Fourier-based moments approach. We present theoretical complexity analysis and numerical experiments to show that our approach can offer significant computational benefits.

## 2   ONLINE SINKHORN

Let $\mathscr{X}$ be a compact subset of $\mathbb{R}^d$ and $\mathscr{C}(\mathscr{X})$ denote the set of continuous functions $\mathscr{X} \to \mathbb{R}$. The Kantorovich formulation was first proposed to study the transport plan between two probability distributions $\alpha, \beta$ with the minimal cost (Kantorovich, 1942):

$$\min_{\pi \in \Pi(\alpha,\beta)} \int C(x,y) \mathrm{d}\pi(x,y), \tag{1}$$

where $\Pi(\alpha, \beta)$ is the set of positive measures with fixed marginals $\alpha$ and $\beta$,

$$\Pi(\alpha,\beta) \overset{\text{def.}}{=} \left\{ \pi \in \mathscr{P}(\mathscr{X}^2) \colon \alpha = \int_{y \in \mathscr{X}} \mathrm{d}\pi(\cdot, y), \ \beta = \int_{x \in \mathscr{X}} \mathrm{d}\pi(x, \cdot) \right\}, \tag{2}$$

and $C \colon \mathscr{X} \times \mathscr{X} \to \mathbb{R}$ is a given cost function. The solution to the optimisation problem equation 1 above can be approximated by a strictly convex optimisation problem by adding a regularisation term: The entropic regularised OT problem with the regularisation parameter $\varepsilon > 0$ is

$$\min_{\pi \in \Pi(\alpha,\beta)} \int C(x,y) \mathrm{d}\pi(x,y) + \varepsilon \mathrm{KL}\left(\pi, \alpha \otimes \beta\right), \tag{3}$$

where $\alpha \otimes \beta$ is the product measure on $\mathscr{X}^2$, and $\mathrm{KL}\left(\pi, \alpha \otimes \beta\right) \overset{\text{def.}}{=} \int \log\left(\frac{\mathrm{d}\pi}{\mathrm{d}\alpha \otimes \beta}\right) \mathrm{d}\pi$ is the Kulback–Leibler divergence (Cuturi, 2013).

The following maximisation problem for $\varepsilon > 0$ is a dual formulation of the entropic regularised OT problem equation 3:

$$F_{\alpha,\beta}(f,g) := \max_{f,g \in \mathscr{C}(\mathscr{X})} \int f(x)d\alpha(x) + \int g(y)d\beta(y) - \varepsilon \int e^{\frac{f \oplus g - C}{\varepsilon}} d\alpha(x)d\beta(y)$$

where $\left(f \oplus g - C\right)(x,y) := f(x) + g(y) - C(x,y)$ and $\left(f, g\right)$ are defined to be the pair of dual potentials (Peyré et al., 2019).

The Sinkhorn algorithm works by alternating minimization on the dual problem $F_{\alpha,\beta}$, and is for discrete distributions. So, to apply Sinkhorn, one first draws empirical distributions $\alpha_n \overset{\text{def.}}{=} \frac{1}{n} \sum_{i=1}^n \delta_{x_i}$ and $\beta_n \overset{\text{def.}}{=} \frac{1}{n} \sum_{j=1}^n \delta_{y_j}$ with $x_i \overset{\text{iid}}{\sim} \alpha$ and $y_i \overset{\text{iid}}{\sim} \beta$, then compute iteratively:

$$u_{t+1} = \left( \frac{1}{n} \sum_{j=1}^n \frac{1}{v_t(y_j)} \exp\left(-\frac{C(x_k,y_j)}{\varepsilon}\right) \right)_{k=1}^n, \quad v_{t+1} = \left( \frac{1}{n} \sum_{i=1}^n \frac{1}{u_{t+1}(x_i)} \exp\left(\frac{-C\left(x_i,y_k\right)}{\varepsilon}\right) \right)_{k=1}^n, \tag{4}$$

where $u_t \overset{\text{def.}}{=} \exp\left(-f_t/\varepsilon\right) \in \mathbb{R}^n$ and $v_t \overset{\text{def.}}{=} \exp\left(-g_t/\varepsilon\right) \in \mathbb{R}^n$ and $f_t, g_t \in \mathbb{R}^n$ are the pair of dual potentials at step $t$. The computational complexity is $O\left(n^2 \log(n) \delta^{-3}\right)$ for reaching $\delta$ accuracy (Altschuler et al., 2017; Mensch & Peyré, 2020).

## 2.1 THE ONLINE SINKHORN ALGORITHM

In the continuous setting, the Sinkhorn iterations operate on functions $u_t, v_t \in \mathcal{C}(\mathcal{X})$ and involve the full distributions $\alpha, \beta$:

$$u_{t+1} = \exp\left(-\frac{f_{t+1}}{\varepsilon}\right) = \int \frac{1}{v_t(y)} K_y(\cdot) d\beta(y), \quad v_{t+1} = \exp\left(-\frac{g_{t+1}}{\varepsilon}\right) = \int \frac{1}{u_{t+1}(x)} K_x(\cdot) d\alpha(x), \quad (5)$$

where $K_y(\cdot) = \exp\left(-\frac{C(\cdot,y)}{\varepsilon}\right)$ and $K_x(\cdot) = \exp\left(-\frac{C(x,\cdot)}{\varepsilon}\right)$. In Mensch & Peyré (2020), a natural extension of the Sinkhorn algorithm was proposed, replacing $\alpha$ and $\beta$ at each step with empirical distributions of growing supports $\hat{\alpha}_t = \frac{1}{n} \sum_{i=n_t}^{n_{t+1}} \delta_{x_i}$ and $\hat{\beta}_t = \frac{1}{n} \sum_{i=n_{t+1}}^{n_{t+1}} \delta_{y_i}$ where $x_i \overset{\text{iid}}{\sim} \alpha$, $y_i \overset{\text{iid}}{\sim} \beta$ and $n_{t+1} = n_t + n$. For appropriate learning rate $\eta_t$, the online Sinkhorn iterations [1] are defined as

$$u_{t+1} = (1 - \eta_t) u_t + \eta_t \int \frac{1}{v_t(y)} K_y(\cdot) d\hat{\beta}_t(y), \quad (6)$$

$$v_{t+1} = (1 - \eta_t) v_t + \eta_t \int \frac{1}{u_{t+1}(x)} K_x(\cdot) d\hat{\alpha}_t(x), \quad (7)$$

where $\eta_t$ is a learning rate to account for the stochastic updates with $\hat{\alpha}_t$ and $\hat{\beta}_t$.

The key observation in Mensch & Peyré (2020) is that, due to equations 6–7, the continuous functions $u_t$ and $v_t$ can be discretely represented using vectors $(q_{i,t}, y_i)$ and $(p_{i,t}, x_i)$, in particular, $u_t$ and $v_t$ take the following form:

$$u_t = \sum_{i=1}^{n_t} \exp\left(\frac{q_{i,t} - C(\cdot, y_i)}{\varepsilon}\right) \quad \text{and} \quad v_t = \sum_{i=1}^{n_t} \exp\left(\frac{p_{i,t} - C(x_i, \cdot)}{\varepsilon}\right), \quad (8)$$

where $q_{i,t}, p_{i,t}$ are weights and $(x_i, y_i)$ are the positions, for $1 \leq i \leq n_t$. Thanks to this representation on $u_t, v_t$, the online Sinkhorn algorithm only needs to record and update vectors $(p_{i,t}, x_i)_i$ and $(q_{i,t}, y_i)_i$. The algorithm is summarised in Algorithm 1.

---

**Algorithm 1:** Online Sinkhorn (Mensch and Peyré, 2020)

---

**Input:** Distributions $\alpha$ and $\beta$, learning weights $(\eta_t)_t$, batch sizes $(b_t)_t$
**Set** $p_i = q_i = 0$ for $i \in (0, n_0]$, where $n_0 = b_0$
**for** $t = 0, \cdots, T - 1$ **do**

  1. Sample $(x_i)_{(n_t, n_{t+1}]} \overset{\text{iid}}{\sim} \alpha$, $(y_i)_{(n_t, n_{t+1}]} \overset{\text{iid}}{\sim} \beta$, where $n_{t+1} = n_t + b_t$
  2. Evaluate $(g_t(y_i))_{i=(n_t, n_{t+1}]}$ via equation 8
  3. $q_{(n_t, n_{t+1}], t+1} \leftarrow \varepsilon \log(\frac{\eta_t}{b_{t+1}}) + (g_t(y_i))_{(n_t, n_{t+1}]}$ and $q_{(0, n_t], t+1} \leftarrow q_{(0, n_t], t} + \varepsilon \log(1 - \eta_t)$
  4. Evaluate $(f_{t+1}(x_i))_{i=(n_t, n_{t+1}]}$ via equation 8
  5. $p_{(n_t, n_{t+1}], t+1} \leftarrow \varepsilon \log(\frac{\eta_t}{b_{t+1}}) + (f_{t+1}(x_i))_{(n_t, n_{t+1}]}$ and $p_{(0, n_t], t+1} \leftarrow p_{(0, n_t], t} + \varepsilon \log(1 - \eta_t)$

**Returns:** $f_T : (q_{i,T}, y_i)_{(0, n_T]}$ and $g_T : (p_{i,T}, x_i)_{(0, n_T]}$

---

Following Mensch & Peyré (2020), convergence can be established under the following three assumptions. The first guarantees well-posedness of the OT problem.

*Assumption* 1. The cost $C : \mathcal{X} \times \mathcal{X} \to \mathbb{R}$ is $L$-Lipschitz continuous for some $L > 0$.

The next two assumptions give conditions on the learning rate $\eta_t$ and sample batch size $b_t$ to gain convergence of the online Sinkhorn algorithm. Assumption 2 is the standard Robins–Monroe stepsize assumption and the 'decaying' stepsize is needed to account for the variance introduced by stochastic updates. Assumption 3 is needed to ensure that the batch size grows sufficiently quickly with respect to the stepsize.

*Assumption* 2. $(\eta_t)_t$ is such that $\sum_{t=1}^{\infty} \eta_t = \infty$ and $\sum_{t=1}^{\infty} \eta_t^2 < \infty$, where $0 < \eta_t < 1$ for all $t > 0$.

*Assumption* 3. $(b_t)_t$ and $(\eta_t)_t$ satisfy that $\sum_{t=1}^{\infty} \frac{\eta_t}{\sqrt{b_t}} < \infty$.

---

[1]In the original Sinkhorn algorithm, the $u_{t+1}$ in equation 7 is replaced with $u_t$, but we consider $u_{t+1}$ in this paper to better match with the classical Sinkhorn algorithm. This makes little difference to the analysis.

In order to satisfy Assumptions 2 and 3, we introduce parameters $a, b$ such that $-1 < b < -\frac{1}{2}$ and $a - b > 1$ and take $\eta_t = t^b$ and $b_t = t^{2a}$.

For the online Sinkhorn algorithm, we obtain the following convergence result:

**Theorem 1.** *Let $f^*$ and $g^*$ denote the optimal potentials. Let $f_t$ and $g_t$ be the output of Algorithm 1 after $t$ iterations. Suppose $\eta_t = t^b$ for $-1 < b < -\frac{1}{2}$, and $b_t = t^{2a}$ with $a - b > 1$. For a constant $c > 0$,*

$$\delta_N \lesssim \exp\left(-cN^{\frac{b+1}{2a+1}}\right) + N^{-\frac{a}{2a+1}} = O\left(N^{-\frac{a}{2a+1}}\right), \tag{9}$$

*where $\delta_N \stackrel{\text{def.}}{=} \left\|\left(f_{t(N)} - f^*\right)/\varepsilon\right\|_{var} + \left\|\left(g_{t(N)} - g^*\right)/\varepsilon\right\|_{var}$, and $t(N)$ is the first iteration such that $\sum_{i=1}^t b_i > N$.*

**Comparison with the previous rate** It was shown in (Mensch & Peyré, 2020, Proposition 4) that

$$\delta_N \lesssim \exp\left(-cN^{\frac{b+1}{2a+1}}\right) + N^{\frac{b}{2a+1}}.$$

By taking $b$ close to $-1$ and $a$ close to $0$, the asymptotic convergence rate can be made arbitrarily close to $O\left(N^{-1}\right)$. There however seems to be an error in the proof (see comment after the sketch proof and the appendix) and $O(N^{-1})$ cannot be achieved. In contrast, our asymptotic convergence rate is $O\left(N^{-\frac{a}{2a+1}}\right)$, which is at best $O(N^{-1/2})$. However, our convergence rate is faster whenever $a \geqslant -b$. See Figure 1.

**Sketch of proof** Let $U_t \stackrel{\text{def.}}{=} (f_t - f^*)/\varepsilon$, $V_t \stackrel{\text{def.}}{=} (g_t - g^*)/\varepsilon$, $e_t \stackrel{\text{def.}}{=} \|U_t\|_{var} + \|V_t\|_{var}$, $\zeta_{\hat{\beta}_t} \stackrel{\text{def.}}{=} \left(T_\beta(g_t) - T_{\hat{\beta}_t}(g_t)\right)/\varepsilon$ and $\iota_{\hat{\alpha}_t} \stackrel{\text{def.}}{=} \left(T_\alpha(f_{t+1}) - T_{\hat{\alpha}_t}(f_{t+1})\right)/\varepsilon$, where $T_\mu$ is defined in Appendix A.2. It can be shown that

$$e_{t+1} \leqslant \left(1 - \eta_t + \eta_t \kappa\right) e_t + \eta_t \left(\|\zeta_{\beta_t}\|_{var} + \|\iota_{\alpha_t}\|_{var}\right),$$

and

$$\mathbb{E}\left\|\zeta_{\hat{\beta}_t}\right\|_{var} \lesssim \frac{A_1(f^*, g^*, \varepsilon)}{\sqrt{b_t}} \quad \text{and} \quad \mathbb{E}\left\|\iota_{\hat{\alpha}_t}\right\|_{var} \lesssim \frac{A_2(f^*, g^*, \varepsilon)}{\sqrt{b_t}},$$

where $\kappa = 1 - \exp(-L\text{diam}(\mathcal{X}))$ using Assumption 1 (Mensch & Peyré, 2020, Lemma 1), and $A_1(f^*, g^*, \varepsilon), A_2(f^*, g^*, \varepsilon)$ are constants depending on $f^*, g^*, \varepsilon$.

Denote $S = A_1(f^*, g^*, \varepsilon) + A_2(f^*, g^*, \varepsilon)$, then taking expectations

$$\mathbb{E}e_{t+1} \lesssim \left(1 - \eta_t + \eta_t \kappa\right) \mathbb{E}e_t + \frac{S\eta_t}{\sqrt{b_t}}. \tag{10}$$

Applying the Gronwall lemma (Mischler, 2018, Lemma 5.1) to this recurrence relation, we get

$$\mathbb{E}e_{t+1} \lesssim \left(\mathbb{E}e_1 + \frac{S}{a - b - 1}\right) \exp\left(\frac{\kappa - 1}{b + 1} t^{b+1}\right) + t^{-a}. \tag{11}$$

The total number of samples $N = O(t^{(2a+1)})$ and this substitution completes the proof. □

In Mensch & Peyré (2020), the last term of the upper bound in equation 11 was calculated as $t^b$, but this followed the derivations of (Moulines & Bach, 2011, Theorem 2) that do not directly apply, which leads to the error in Theorem 1 to be $O(N^{\frac{b}{2a+1}})$.

To show that the error rates in Theorem 1 are correct, we plot the variational error for online Sinkhorn and display our theoretical convergence rate with exponent $\frac{-a}{2a+1}$ and the 'old theoretical' convergence rate with exponent $\frac{b}{2a+1}$ from (Mensch & Peyré, 2020, Proposition 4). Our experiments are in 1D, 2D and 5D. For the 1D case, the source and target distributions are the Gaussian distributions $\mathcal{N}(3, 4)$ and $\mathcal{N}(1, 2)$. In 2D, the source distribution $\mathcal{N}\left(\mu^{(1)}, \Sigma^{(1)}\right)$ is generated by $\mu_i^{(1)} \sim \mathcal{U}(0, 10)$, for $i = 1, 2$ and $\Sigma^{(1)}$ is a randomly generated covariance matrix. The target distribution $\mathcal{N}\left(\mu^{(2)}, \Sigma^{(2)}\right)$ is generated by $\mu_i^{(2)} \sim \mathcal{U}(0, 5)$, for $i = 1, 2$ and $\Sigma^{(2)}$ is a randomly generated covariance matrix. The $y$-axis in all the plots in this section are the errors of the potential functions $\|f_t - f_{t-1}\|_{var} + \|g_t - g_{t-1}\|_{var}$ based on a finite set of points $x_i, y_i$. The plots show that the convergence behaviour of online Sinkhorn asymptotically is parallel to our theoretical line, which corroborates with Theorem 1.

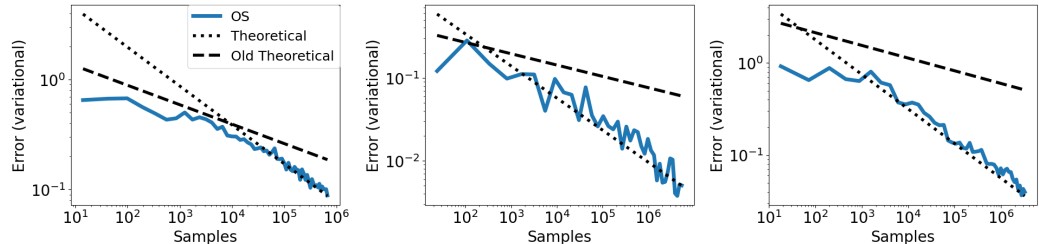

Figure 1: Left: $\varepsilon = 0.3$, $a = 1.2$, $b = -0.6$, $d = 1$. The theoretical rate of convergence is $-0.35$ (compared to the old rate of $-0.18$); a linear fit shows OS is converging with rate $-0.31$. Middle: $\varepsilon = 0.3$, $a = 1.7$, $b = -0.6$, $d = 2$ with rates `Theoretical` $= -0.39$; `Old theoretical` $= -0.14$; `OS` $= -0.37$. Right: $\varepsilon = 0.4$, $a = 1.5$, $b = -0.55$, $d = 5$ with rates `Theoretical` $= -0.38$; `Old theoretical` $= -0.14$; `OS` $= -0.41$.

## 3 COMPRESSED ONLINE SINKHORN

To motivate this section, we first discuss the computational complexity of online Sinkhorn and contrast it with that of classical Sinkhorn: At each iteration, the computational complexity of online Sinkhorn is $O(n_t b_t) = O(t^{4a+1})$ and the memory cost for $(x_i, y_i)_{i=(0,n_t]}$ is $O(n_t) = O(t^{2a+1})$. Due to the increasing batchsize $b_t$, these costs increase polynomially with each iteration. Note also that in Theorem 1, the total number of samples $N = O(t^{2a+1})$ and one requires a computational cost of $O(N^2)$ to achieve the error bound equation 9. The other direct strategy is to simply draw $N$ iid samples from $\alpha$ and $\beta$ and apply Sinkhorn's algorithm directly; the analogous complexity statement for Sinkhorn is, for some $\lambda > 0$,

$$\|f_t - f^*\|_{\text{var}} + \|g_t - g^*\|_{\text{var}} \leq O(\lambda^t + N^{-1/2}) \tag{12}$$

with the computational complexity of achieving this error bound being $O(tN^2)$. This is due to the fact that the per-iteration complexity of Sinkhorn is $O(N^2)$; the Sinkhorn algorithm converges linearly at rate $O(\lambda^t)$ for some $\lambda \in (0, 1)$, where $t$ is the iteration (Franklin & Lorenz (1989); Peyré et al. (2019)) (hence yielding the first term in equation 12); and the sampling error in approximating the true Kantorovich potentials with those computed from the empirical measures is $O(N^{-1/2})$ Rigollet & Stromme (2022) (yielding the second term in equation 12). From this comparison, one can expect online Sinkhorn to provide advantages initially, but the asymptotic performance is still the same (obtaining $O(N^{-1/2})$ convergence in the potentials with $O(N^2)$ complexity). In this section, we will introduce a compressed version of online Sinkhorn: the polynomial growth in the per-iteration complexity in online Sinkhorn is due to the increasing sizes in the discrete representations of $u_t$ $v_t$, to mitigate this and hence improve the asymptotic convergence behaviour, we will compress these representations using measure compression techniques.

We now derive the *compressed online Sinkhorn* algorithm. To explain our derivation, we focus on obtaining a compressed version of $u_t$; the function $v_t$ can be treated in a similar manner. From equation 8, we have $u_t(x) = \int \frac{K_x(y)}{\varphi(y)} \mathrm{d}\mu(y)$ for $\mu \stackrel{\text{def.}}{=} \sum_{i=1}^{n_t} \exp\left(\frac{q_{i,t}}{\varepsilon}\right) \varphi(y_i) \delta_{y_i}$, $K_x(y) \stackrel{\text{def.}}{=} \exp(-C(x, y)/\varepsilon)$ and any positive function $\varphi$. The idea of our compression method is to exploit measure compression techniques to replace $\mu$ with a measure $\hat{\mu}$ made up of $m \ll n_t$ Diracs. We then approximate $u_t$ with $\hat{u}_t(x) = \int \frac{K_x(y)}{\varphi(y)} \mathrm{d}\hat{\mu}(y)$. Let $\varphi = v_t = \exp\left(-\frac{g_t}{\varepsilon}\right)$, then $\exp\left(\frac{q_{i,t}}{\varepsilon}\right) \varphi(y_i) \leq 1$ by the update 3 Algorithm 1. The weights for $\mu$ are hence bounded and it is easier to compress $\mu$. To ensure that $\hat{u}_t \approx u_t$, we will enforce that the measure $\hat{\mu}$ satisfies

$$\int P_k(y) \mathrm{d}\mu(y) = \int P_k(y) \mathrm{d}\hat{\mu}(y) \tag{13}$$

for some appropriate set of functions $\{P_k ; k \in \Omega\}$, where $\Omega$ is some set of cardinality $m_t$. Let us introduce a couple of ways to choose such functions.

**Example: Gaussian quadrature** In dimension 1 (covered here mostly for pedagogical purposes), one can consider the set of polynomials up to degree $2m - 1$, denoted by $\mathbb{P}_{2m-1}$ for some $m \in \mathbb{N}$. For well-ordered $y_i$, the constraints (13) defines the $m$-point Gaussian quadrature $\hat{\mu} = \sum \hat{w}_i \delta_{\hat{y}_i}$, and both new weights $\hat{w}_i$ and new nodes $\hat{y}_i$ are required to achieve this. Numerically, this can be efficiently implemented following the `OPQ` Matlab package (Gautschi, 2004) and the total

complexity for the compression is $O(m^3 + n_t m)$. To define $\hat{u}_t$, solve $e^{\hat{q}_{i,t}/\varepsilon} v_t(\hat{y}_i) = \hat{w}_i$ for $\hat{q}_{i,t}$ and let $\hat{u}_t(x) = \sum_{i=1}^{m} \exp(\hat{q}_{i,t}/\varepsilon) K_x(\hat{y}_i)$.

---

**Algorithm 2:** Compressed online Sinkhorn

---

**Input:** Distributions $\alpha$ and $\beta$, learning rates $(\eta_t)_t = ((t+1)^b)_t$, batch size $(b_t)_t = ((t+1)^{2a})_t$, such that $-1 < b < -\frac{1}{2}$ and $a - b > 1$

**Initialisation:** $n_1 = m_1 = b_1, \hat{p}_{i,1} = \hat{q}_{i,1} = 0, \hat{x}_{i,1} \overset{\text{iid}}{\sim} \alpha, \hat{y}_{i,1} \overset{\text{iid}}{\sim} \beta$ for $i \in (0, m_1]$

**for** $t = 1, \cdots, T-1$ **do**

> 1. Sample $(x_i)_{(n_t, n_{t+1}]} \overset{\text{iid}}{\sim} \alpha$, $(y_i)_{(n_t, n_{t+1}]} \overset{\text{iid}}{\sim} \beta$, where $n_{t+1} = n_t + b_t$. Let $(\hat{x}_{i,t})_{(m_t, m_t+b_t]} = (x_i)_{(n_t, n_{t+1}]}, (\hat{y}_{i,t})_{(m_t, m_t+b_t]} = (y_i)_{(n_t, n_{t+1}]}$.
>
> 2. Evaluate $(\hat{g}_t(y_i))_{i=(n_t, n_{t+1}]}$, where $\hat{g}_t = -\varepsilon \log \sum_{i=1}^{m_t} \exp(\frac{\hat{p}_{i,t} - C(\hat{x}_{i,t}, \cdot)}{\varepsilon})$.
>
> 3. $q_{(m_t, m_t+b_t], t+1} \leftarrow \varepsilon \log\left(\frac{\eta_t}{b_t}\right) + (\hat{g}_t(y_i))_{(n_t, n_{t+1}]}$ and $q_{(0, m_t], t+1} \leftarrow \hat{q}_{(0, m_t], t} + \varepsilon \log(1 - \eta_t)$.
>
> 4. Evaluate $(f_{t+1}(x_i))_{i=(n_t, n_{t+1}]}$, where $f_{t+1} = -\varepsilon \log\left(\sum_{i=1}^{m_t+b_t} \exp\left(\frac{q_{i,t+1} - C(\cdot, \hat{y}_{i,t})}{\varepsilon}\right)\right)$
>
> 5. $p_{(m_t, m_t+b_t], t+1} \leftarrow \varepsilon \log\left(\frac{\eta_t}{b_t}\right) + (f_{t+1}(x_i))_{(n_t, n_{t+1}]}$ and $p_{(0, m_t], t+1} \leftarrow \hat{p}_{(0, m_t], t} + \varepsilon \log(1 - \eta_t)$.
>
> 6. Compression: find $\hat{u}_{t+1} = e^{-\hat{f}_{t+1}}$ to approximate $u_{t+1} = e^{-f_{t+1}/\varepsilon}$ such that equation 13 holds (for $\mu, \hat{\mu}$ corresponding to $u_t, \hat{u}_t$) with $m_t$ points. Define $\hat{y}_{i,t+1}, \hat{q}_{i,t+1}$. so that $\hat{f}_{t+1} = -\varepsilon \log \sum_{i=1}^{m_t+1} \exp((\hat{q}_{i,t+1} - C(\hat{y}_{i,t+1}, \cdot))/\varepsilon)$,
>
> 7. Similarly update $\hat{y}_{i,t+1}$ and $\hat{p}_{i,t+1}$.

**Returns:** $\hat{f}_T = -\varepsilon \log \sum_{i=1}^{m_T} \exp(\frac{\hat{q}_{i,T} - C(\cdot, \hat{y}_{i,T})}{\varepsilon}), \hat{g}_T = -\varepsilon \log \sum_{i=1}^{m_t} \exp(\frac{\hat{p}_{i,T} - C(\hat{x}_{i,T}, \cdot)}{\varepsilon})$

---

**Example: Fourier moments** Since we expect that the compressed function $\hat{u}_t$ satisfies $\hat{u}_t \approx u_t$, one approach is to ensure that the Fourier moments of $\hat{u}_t$ and $u_t$ match on some set of frequencies. Observe that the Fourier transform of $u_t(x) = \int \frac{K_x}{v_t}(y) \, d\mu(y)$ can be written as

$$\int \exp(-ikx) \int \frac{K_x}{v_t}(y) \, d\mu(y) \, dx = \int \frac{\hat{K}_y(k)}{v_t(y)} \, d\mu(y), \quad \text{where} \quad \hat{K}_y(k) \overset{\text{def.}}{=} \int K_x(y) \exp(-ikx) \, dx. \tag{14}$$

We therefore let $P_k(y) = \frac{\hat{K}_y(k)}{v_t(y)}$ in equation 13, for suitably chosen frequencies $k$ (see Appendix A.5.2).

In contrast to GQ, we retain the current nodes $y_i$ and seek new weights $\hat{w}_i$ such that $\hat{\mu} = \sum_{i=1}^{m} \hat{w}_i \delta_{y_i}$ satisfies $\int \frac{\hat{K}_y(k)}{v_t(y)} \, d(\mu - \hat{\mu})(y) = 0$. Let $b_t = (\int P_k(y) d\mu(y))$ and $M_t = (P_k(y_i))_{k \in \Omega, i \in [n_t]}$; equivalently, we seek $\hat{w} \geq 0$ such that $M_t \hat{w} = b_t$. We know from the Caratheodory theorem that there exists a solution with only $m+1$ positive entries in $\hat{w}$, and the resulting compressed measure $\hat{\mu} = \sum_{\hat{w}_i > 0} \hat{w}_i \delta_{\hat{y}_i}$. Algorithms to find the Caratheodory solution include Hayakawa et al. (2022) and Tchernychova (2015). We may also find $\hat{w} \geq 0$ to minimise $\|M_t \hat{w} - b_t\|^2$ using algorithms such as Virtanen et al. (2020) and Pedregosa et al. (2011), which in practice is faster but leads to a sub-optimal support: in our experiments, these methods contain up to $10m$ points. Theoretically, as long as the size of the support is $O(m)$, the same behaviour convergence guarantees hold, so although the non-negative least squares solvers do not provide this guarantee, we find it to be efficient and simple to implement in practice. The complexity of calculating the matrix $M_t$ is $O(dmn_t)$, and for solving the linear system is $O(m^3)$ or $O(m^3 \log n_t)$ depending on the solution method. Therefore, the total complexity for the compression method is $O(m^3 + dmn_t)$ or $O(m^3 \log n + dmn_t)$. In practice, we found the non-negative least squares solvers to be faster and to give good accuracy. The compressed online Sinkhorn algorithm is summarised in Algorithm 2. The main difference is the Steps 6 and 7, where the further compression steps are taken after the online Sinkhorn updates in Algorithm 1. This reduces the complexity of the evaluation of $(f_t, g_t)$ in the next iteration.

### 3.1 COMPLEXITY ANALYSIS FOR COMPRESSED ONLINE SINKHORN

Let us return to our examples from Section 3 and look at the error incurred by Gaussian quadrature and Fourier moments compression. The details can be found in Appendix A.5.

**Definition 2.** *The compression algorithm has rate $\zeta > 0$ with compression constant $C_d > 0$ if the compression of $u_t$ to the $m$-term $\hat{u}_t$ satisfies $|u(x) - \hat{u}_t(x)| \le C_d / m^\zeta$ for any $x \in \mathcal{X}$.*

**Gaussian quadrature** Note that $K_x$ and $v_t$ are smooth (both have the same regularity as $e^{-C(x,y)/\varepsilon}$; see equation 8). By (Gautschi, 2004, Corollary to Theorem 1.48), the compression error by Gaussian quadrature is

$$u_t(x) - \hat{u}_t(x) = \int \frac{K_x}{v_t}(y) \mathrm{d}\mu(y) - \int \frac{K_x}{v_t}(y) \mathrm{d}\hat{\mu}(y) = O\left(\frac{1}{\varepsilon^m m!}\right).$$

In particular, though limited to one dimension, GQ converges exponentially and hence with rate $\zeta$ for any $\zeta > 0$ as long as the derivatives of $u$ and $v$ are uniformly bounded.

**Fourier moments** We show in Appendix A.5 that, for $C(x,y) = \left\| x - y \right\|^2$,

$$u_t(x) - \hat{u}_t(x) = \int \frac{K_x}{v_t}(y) d\mu(y) - \int \frac{K_x}{v_t}(y) d\hat{\mu}(y) = \int \frac{\varphi_x(z)}{v_t(y)} \mathrm{d}z, \tag{15}$$

where

$$\varphi_x(k) = \exp(i z k) \sqrt{\varepsilon\pi} \exp\left(-\frac{\varepsilon k^2}{4}\right) \int \exp(-i k y) \mathrm{d}(\mu - \hat{\mu})(y). \tag{16}$$

By choice of $P_k$ and equation 13, $\varphi_x(k) = 0$ for $k \in \Omega$ and the compression error via Gaussian QMC sampling (Kuo & Nuyens, 2016) is $O(|\log m|^d|/m)$. This is much slower than GQ and converges with rate $\zeta$ for any $\zeta < 1$. For each dimension $d$, the rate $\zeta < 1$ holds asymptotically though the constant $C_d \to \infty$ as $d \to \infty$ (see Definition 2).

We now present the convergence theorem for Algorithm 2. To ensure the compression error is consistent with the Online Sinkhorn error, we assume the following.

*Assumption* 4. Assume that at step $t$

$$\sup_x \left| f_t(x) - \hat{f}_t(x) \right| = O\left(t^{b-a}\right) \quad \text{and} \quad \sup_y \left| g_t(y) - \hat{g}_t(y) \right| = O\left(t^{b-a}\right).$$

As $f_t(x)$ is a Lipschitz function of $u_t(x)$, both $u_t, f_t$ have the same compression rate $\zeta$ and Assumption 4 is equivalent to choosing $m_t$ so that $C_d / m_t^\zeta = t^{b-a}$. For a compression method, we need to choose appropriate values for $C_d$ and $\zeta$ in Defintion 2 to determine the compression size $m_t$.

**Theorem 3.** *Let $f^*$ and $g^*$ denote the optimal potentials. Let $\hat{f}_t$ and $\hat{g}_t$ be the output of Algorithm 2 after $t$ iterations. Under Assumptions 1 to 4, $\eta_t = t^b$ for $-1 < b < -\frac{1}{2}$, and $b_t = t^{2a}$ with $a - b > 1$, we have*

$$\hat{\delta}_N \lesssim \exp\left(-cN^{\frac{b+1}{2a+1}}\right) + N^{-\frac{a}{2a+1}} = O\left(N^{-\frac{a}{2a+1}}\right), \tag{17}$$

*where $\hat{\delta}_N = \left\| (\hat{f}_{t(N)} - f^*) / \varepsilon \right\|_{var} + \left\| (\hat{g}_{t(N)} - g^*) / \varepsilon \right\|_{var}$, $t(N)$ is the first iteration number for which $\sum_{i=1}^t b_i > N$, and $c$ is a positive constant.*

**Sketch of proof** The proof follows the proof of Theorem 1 with the extra compression errors. Further to the definitions of $U_t$ and $V_t$, define the following terms for $\hat{f}_t$ and $\hat{g}_t$:

$$\hat{U}_t \stackrel{\text{def.}}{=} (\hat{f}_t - f^*) / \varepsilon, \qquad\qquad \hat{V}_t \stackrel{\text{def.}}{=} (\hat{g}_t - g^*) / \varepsilon, \tag{18}$$

$$\hat{\zeta}_{\hat{\beta}_t} \stackrel{\text{def.}}{=} \left(T_\beta(\hat{g}_t) - T_{\hat{\beta}_t}(\hat{g}_t)\right) / \varepsilon, \qquad\qquad \hat{\imath}_{\hat{\alpha}_t} \stackrel{\text{def.}}{=} \left(T_\alpha(f_{t+1}) - T_{\hat{\alpha}_t}(f_{t+1})\right) / \varepsilon. \tag{19}$$

Then we have the following relation

$$\hat{U}_{t+1} = (\hat{f}_{t+1} - f_{t+1}) / \varepsilon + (f_{t+1} - f^*) / \varepsilon \stackrel{\text{def.}}{=} \mathrm{err}_{f_{t+1}} + U_{t+1},$$

$$\hat{V}_{t+1} = (\hat{g}_{t+1} - g_{t+1}) / \varepsilon + (g_{t+1} - g^*) / \varepsilon \stackrel{\text{def.}}{=} \mathrm{err}_{g_{t+1}} + V_{t+1},$$

where $\text{err}_{f_{t+1}} = (\hat{f}_{t+1} - f_{t+1})/\varepsilon$ and $\text{err}_{g_{t+1}} = (\hat{g}_{t+1} - g_{t+1})/\varepsilon$. Notice that $\|\text{err}_{f_{t+1}}\|_{var} = O(t^{-a+b})$ under Assumption 4.

Define $\hat{e}_t \stackrel{\text{def.}}{=} \|\hat{U}_t\|_{var} + \|\hat{V}_t\|_{var}$, then for $t$ large enough

$$\hat{e}_{t+1} \leqslant (1 - \eta_t + \eta_t \kappa) \hat{e}_t + \eta_t \left( \|\hat{\zeta}_{\hat{\beta}_t}\|_{var} + \|\hat{\iota}_{\hat{\alpha}_t}\|_{var} \right) + \|\text{err}_{f_{t+1}}\|_{var} + \|\text{err}_{g_{t+1}}\|_{var}. \tag{20}$$

Similarly to the proof of Theorem 1, we can show that

$$\mathbb{E}\left\|\zeta_{\hat{\beta}_t}\right\|_{var} \lesssim \frac{A_1(f^*, g^*, \varepsilon)}{\sqrt{b_t}} \quad \text{and} \quad \mathbb{E}\left\|\iota_{\hat{\alpha}_t}\right\|_{var} \lesssim \frac{A_2(f^*, g^*, \varepsilon)}{\sqrt{b_t}},$$

where $A_1(f^*, g^*, \varepsilon), A_2(f^*, g^*, \varepsilon)$ are constants depending on $f^*, g^*, \varepsilon$. Taking expectations and applying the Gronwall lemma (Mischler, 2018, Lemma 5.1) to the recurrence relation, we have

$$\mathbb{E}\hat{e}_{t+1} \lesssim \left( \mathbb{E}e_1 + \frac{1}{a - b - 1} \right) \exp\left( \frac{\kappa - 1}{b + 1} t^{b+1} \right) + \frac{1}{(1 - \kappa)} t^{-a}. \qquad \square$$

To reach the best asymptotic behaviour of the convergence rate $N^{-\frac{a}{2a+1}}$ in Theorem 3, we want $a \to \infty$. In this limit however, the transient behaviour $\exp(-cN^{\frac{b+1}{2a+1}})$ becomes poorer and, in practice, a moderate value of $a$ must be chosen.

We now look at the complexity of the algorithms in terms of the compression rate $\zeta$.

**Proposition 4.** *Under Assumption 4, we further assume that compressing a measure from n atoms to m has complexity $C(n, m) = O(m^3 + nm)$ and that $m_t = t^{\frac{a-b}{\zeta}}$ at step $t$. The computational complexity of reaching accuracy $\mathbb{E}e_t \leqslant \delta$ for Algorithm 2 is*

$$\hat{\mathscr{C}} = O\left( \delta^{-\left(2 + \frac{a-b}{a\zeta} + \frac{1}{a}\right)} \right), \quad if \zeta \geqslant \frac{a-b}{a} \quad and \quad \hat{\mathscr{C}} = O\left( \delta^{-\left(\frac{3(a-b)}{a\zeta} + \frac{1}{a}\right)} \right), \quad if \zeta < \frac{a-b}{a}. \tag{21}$$

*The complexity of reaching the same accuracy $\delta$ for online Sinkhorn is $\mathscr{C} = O\left( \delta^{-\left(4 + \frac{2}{a}\right)} \right)$.*

We want to choose $m_t$ such that the extra compression cost $C(n_t, m_t)$ does not exceed the online Sinkhorn complexity $O(n_t b_t)$ at step $t$. The ratio of the complexities for the compressed and original online Sinkhorn algorithm is

$$\frac{\hat{\mathscr{C}}}{\mathscr{C}} = O\left( \delta^{2 + \frac{1}{a} - \frac{a-b}{a\zeta}} \right), \quad \text{if } \zeta \geqslant \frac{a-b}{a} \quad \text{and} \quad \frac{\hat{\mathscr{C}}}{\mathscr{C}} = O\left( \delta^{4 + \frac{1}{a} - \frac{3(a-b)}{a\zeta}} \right), \quad \text{if } \zeta < \frac{a-b}{a}. \tag{22}$$

When $\zeta > \frac{3(a-b)}{4a+1}$, the exponent is positive and the compressed Online Sinkhorn is more efficient. The larger this exponent, the more improvement we can see in the asymptotical convergence of the compressed online Sinkhorn compared to the online Sinkhorn. These gains concern the asymptotic rate and, in practice, it is hard to realise these gains due to transient behaviour and the size of the compression constant $C_d$.

## 4 NUMERICAL EXPERIMENTS

In Figure 2(a)–(c), we compare online Sinkhorn (OS) with compressed online Sinkhorn (COS) with Fourier compression in 1D, 2D and 5D, and with Gauss quadrature (GQ) in 1D. Recall that the learning rate is $\eta_t = t^b$ and the batch size at step $t$ is $b_t = t^{2a}$, and the compression is determined by $C_d$ and $\zeta$ (see Definition 2). We chose different sets of parameters for the experiments (a)–(c) and note run times. The run times measure the time to complete the iterations on the lower plot (so the relative errors in the objective are matched).

| | $d$ | $\varepsilon$ | $a$ | $b$ | $\zeta$ | $C_d$ | Runtime | OS | COS-Fourier | COS-GQ |
|---|---|---|---|---|---|---|---|---|---|---|
| (a) | 1 | 0.4 | 1.5 | -0.6 | 2 | 1 | | 130s | 130s | 151s |
| (b) | 2 | 0.5 | 1.2 | -0.6 | 0.9 | 1 | | 449s | 100s | - |
| (c) | 5 | 0.5 | 1.2 | -0.6 | 0.9 | 3 | | 153s | 92s | - |

To motivate the choice of parameters, recall from Theorem 3 that the transient error contribution behaves like $\exp(-cN^{\frac{b+1}{2a+1}})$ and decays more quickly when $a$ is small (its minimum value is $1 + b$)

and $b$ is large (its maximum value is $-0.5$); in contrast, the asymptotic error contribution behaves like $N^{-\frac{a}{2a+1}}$ and decays faster when $a$ is large. In our experiments, we choose $b = -0.6$, and $a$ between 1.2 and $a = 1.5$ as a compromise in order to focus on the asymptotic rate. The compression methods are applied once the total sample size reaches 1000. In (a), we use the same Gaussian distribution as in Section 2. For (b) and (c), we used a Gaussian Mixture Model (GMM) as described in Appendix A.1. We display the errors of the potential functions $\|f_t - f_{t-1}\|_{var} + \|g_t - g_{t-1}\|_{var}$ (for a variation distance based on a finite set of points) and the relative errors of the objective functions $F(f_t, g_t) - F(f^*, g^*)$. For computing relative errors, the exact value is explicitly calculated for the Gaussian example (a) following (Janati et al., 2020). For the Gaussian Mixture Models (b) and (c), the reference value is taken from online Sinkhorn with approximately $N = 10^5$ samples. For the parameters choice in Figure 2, in the 1D experiments, the complexities of GQ and Fourier COS are $\hat{\mathscr{C}} = O(\delta^{-\frac{101}{30}})$ and $\hat{\mathscr{C}} = O(\delta^{-\frac{290}{57}})$ respectively, with the complexity of OS being $\mathscr{C} = O(\delta^{-\frac{16}{3}})$. In 2 and 5D experiments, the complexities for the Fourier COS are $\hat{\mathscr{C}} = O(\delta^{-\frac{35}{6}})$, where the complexity for OS is $\mathscr{C} = O(\delta^{-\frac{17}{3}})$. As shown in Figure 2, even when the choice of $\zeta$ is outside the optimal range in $2D$, we still observe a better running time.

The numerical simulations were implemented in Python and run using the NVIDIA Tesla K80 GPU on Colab. To reduce the memory cost, we employed the KeOps package (Charlier et al., 2021) calculating the $C$-transforms in Algorithm 1. Execution times are shown and demonstrate the advantage of the compression method. Here the compression is computed using Scipy's `NNLS`.

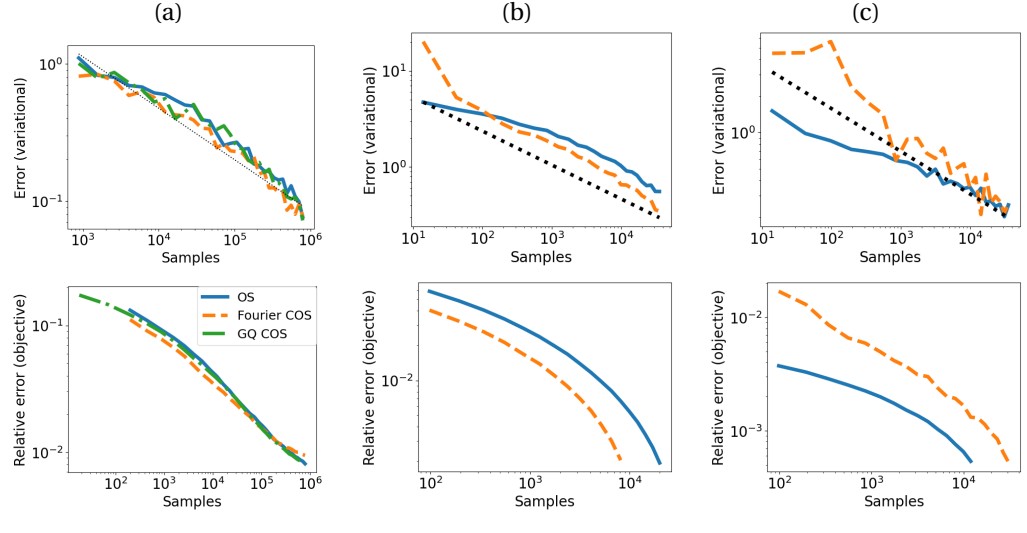

Figure 2:

## 5 CONCLUSION AND FUTURE WORK

In this paper, we revisited the online Sinkhorn algorithm and provided an updated convergence analysis. We also proposed to combine this algorithm with measure compression techniques and demonstrated both theoretical and practical gains in some low-dimensional problems. We focused on the use of Fourier moments as a compression technique, and this worked particularly well with quadratic costs due to the fast tail behaviour of $\exp(-x^2)$. However, Fourier compression generally performs less well for non-quadratic costs (such as $C(x, y) = \|x - y\|_1$) due to the slower frequency decay of $\exp(-C)$. We leave investigations into compression of such losses with slow tail behaviour as future work. In high dimensions, the technique is limited by suitable compression algorithms and, in the proposed QMC-based algorithm, there is a logarithmic dependence on the error that limits gains in numerical experiments for $d \geq 5$. Another direction of future work is to further investigate the non-asymptotic convergence behaviour: In Theorem 1, there is a linear convergence term which dominates in the non-asymptotic regime and it could be beneficial to adaptively choose the parameters $a, b$ to exploit this fast initial behaviour.

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

## A APPENDIX

### A.1 GAUSSIAN MIXTURE MODEL

For experiments (b) and (c) in Figure 2, we set up test distributions $\alpha, \beta$ that sample from $\mathcal{N}(\mu_i, \Sigma_i)$ for $i = 1, 2$ with equal probability. For (b), the entries of $\mu_1$ are iid samples from $\mathcal{N}(0, 100)$ for $\alpha$, and from $\mathcal{N}(0, 25)$ for $\beta$; we take $\mu_2 = -\mu_1$ in both cases. The covariance matrices $\Sigma_i = c_i Q Q^T$ for $Q$ with iid standard Gaussian entries using $c_1 = 3$ and $c_2 = 4$. For (c), the same set-up is used with $c_1 = 1$ and $c_2 = 1$.

### A.2 USEFUL LEMMAS

Define the operator $T_\mu$ to be Mensch & Peyré (2020)

$$T_\mu(h) = -\varepsilon \log \int_{y \in \mathcal{X}} \exp\left(h(y) - C(\cdot, y)\right) d\mu(y), \tag{23}$$

then the updates with respect to the potentials $(f_t, g_t)$ are

$$f_{t+1} = T_\beta(g_t) \quad \text{and} \quad g_{t+1} = T_\alpha(f_t). \tag{24}$$

Under Assumption 1, a soft $C$-transform is always Lipschitz, and the following result can be found in (Vialard, 2019, Proposition 15).

**Lemma 5.** *Under Assumption 1, a soft C-transform $f = T_\mu(g)$ with a probability measure $\mu$ is $\varepsilon L$-Lipschitz, where L is the Lipschitz constant of C defined in Assumption 1.*

We also show that the error $\sup_{x \in \mathcal{X}} |f_t - f^*|, \sup_{x \in \mathcal{X}} |g_t - g^*|$ in the online Sinkhorn is uniformly bounded.

**Lemma 6.** *Suppose for some $t > 0$, $\max\left\{\sup\limits_{x \in \mathcal{X}} |f_t - f^*| < \delta, \sup\limits_{y \in \mathcal{X}} |g_t - g^*|\right\} < \delta$, where $(f^*, g^*)$ is the pair of optimal potentials. Let $(f_{t+1}, g_{t+1})$ be the pair of potentials in the following updates*

$$\exp\left(-f_{t+1}(x)/\varepsilon\right) = (1 - \eta_t)\exp\left(-f_t(x)/\varepsilon\right) + \eta_t \int \exp\left((g_t(y) - C(x,y))/\varepsilon\right) \mathrm{d}\hat{\beta}_t(y), \quad (25)$$

$$\exp\left(-g_{t+1}(x)/\varepsilon\right) = (1 - \eta_t)\exp\left(-g_t(x)/\varepsilon\right) + \eta_t \int \exp\left((f_{t+1}(y) - C(x,y))/\varepsilon\right) \mathrm{d}\hat{\alpha}_t(x), \quad (26)$$

*where $\hat{\alpha}_t, \hat{\beta}_t$ are two probability measures. Then $\sup\limits_{x \in \mathcal{X}} |f_{t+1} - f^*| < \delta$.*

*Proof.* Multiply by $\exp\left(f^*/\varepsilon\right)$ on both sides of the equation 25,

$$\exp(f^*/\varepsilon - f_{t+1}/\varepsilon) = (1 - \eta_t)\exp(f^*/\varepsilon - f_t/\varepsilon) + \eta_t \int \exp((f^* + g_t - C)/\varepsilon)\mathrm{d}\hat{\beta}_t,$$

$$= (1 - \eta_t)\exp(f^*/\varepsilon - f_t/\varepsilon) + \eta_t \int \exp((f^* + g^* - g^* + g_t - C)/\varepsilon)\mathrm{d}\hat{\beta}_t,$$

$$= (1 - \eta_t)\exp(f^*/\varepsilon - f_t/\varepsilon) + \eta_t \int \exp((g_t - g^*)/\varepsilon)\mathrm{d}\hat{\beta}_t$$

$$< (1 - \eta_t)\exp(\delta/\varepsilon) + \eta_t \int \exp(\delta/\varepsilon)\mathrm{d}\hat{\beta}_t = \exp(\delta/\varepsilon).$$

Take logs on both sides to get $\sup\limits_{x \in \mathcal{X}} |f^* - f_{t+1}| < \delta$. A similar argument gives the lower bound and we see that $\sup\limits_{x \in \mathcal{X}} |f^* - f_{t+1}| < \delta$. $\qquad\square$

As an example of (Van der Vaart, 2000, Chapter 19), we have the following results for function $\varphi(x) = \int \exp\left((f(x) + g(y) - C(x,y))/\varepsilon\right)\mathrm{d}\beta(y)$.

**Lemma 7.** *Under Assumption 1, let $y_1, \cdots, y_n$ be i.i.d. random samples from a probability distribution $\beta$ on $\mathcal{X}$, and define for Lipschitz functions $f$ and $g$,*

$$\varphi_i(x) = \exp\left((f(x) + g(y_i) - C(x, y_i))/\varepsilon\right),$$

$$\varphi(x) = \int \exp\left((f(x) + g(y) - C(x, y))/\varepsilon\right)\mathrm{d}\beta(y). \quad (27)$$

*Suppose that $\left\|\exp\left((f(x) + g(y) - C(x,y))/\varepsilon\right)\right\|_\infty \leq M$ for some $M > 0$, where $\|\cdot\|_\infty$ is the supremum norm. Then there exists $A > 0$ depending on $f, g, \varepsilon$ such that, for all $n > 0$,*

$$\mathbb{E}\sup_{x \in \mathcal{X}}\left|\frac{1}{n}\sum_{i=1}^{n}\varphi_i(x) - \varphi(x)\right| \leq \frac{A}{\sqrt{n}}.$$

*Moreover, $\sup\limits_{x \in \mathcal{X}}\left|\frac{1}{n}\sum_{i=1}^{n}\varphi_i(x) - \varphi(x)\right| \xrightarrow{a.s.} 0$ as $n \to \infty$.*

Make use of Lemma 7 to further prove the difference of two $T_\beta(g)$ and $T_{\hat{\beta}}(g)$ has an upper bound in $\|\cdot\|_{var}$.

**Lemma 8.** *Suppose that $g$ is Lipshitcz and $(f^*, g^*)$ is the pair of optimal potentials. Denote $\zeta_{\hat{\beta}}(x) \stackrel{\text{def.}}{=} \frac{\left(T_\beta(g) - T_{\hat{\beta}}(g)\right)}{\varepsilon}(x)$, where $\hat{\beta} = \frac{1}{n}\sum_{i=1}^{n}\delta_{y_i}$ is an empirical probability measure with $y_i \stackrel{\text{iid}}{\sim} \beta$, $n \in \mathbb{Z}_+$, and $M(x) = \int_y \exp\left((f^*(x) + g(y) - C(x,y))/\varepsilon\right)\mathrm{d}\beta(y)$ is bounded from above and below with respect to $x$.*

*Then, there exists a constant $A_1(f^*, g^*, \varepsilon)$ depending on $f^*, g^*, \varepsilon$, such that*

$$\mathbb{E}\|\zeta_{\hat{\beta}}\|_{var} \lesssim \frac{A_1(f^*, g^*, \varepsilon)}{\sqrt{n}}. \quad (28)$$

*Proof.* For all $x \in \mathcal{X}$, using that $f^* = T_\beta(g^*)$,

$$\left\|\zeta_{\hat{\beta}}\right\|_{var} = \left\|\left(T_\beta(g) - T_{\hat{\beta}}(g)\right)/\varepsilon\right\|_{var} = \left\|\log\left(\frac{\exp\left(-T_{\hat{\beta}}(g)/\varepsilon\right)}{\exp\left(-T_\beta(g)/\varepsilon\right)}\right)\right\|_{var}$$

$$= \left\|\log\left(1 + \frac{\exp\left(-T_{\hat{\beta}}(g)/\varepsilon\right) - \exp\left(-T_\beta(g)/\varepsilon\right)}{\exp\left(-T_\beta(g)/\varepsilon\right)}\right)\right\|_{var}$$

$$= \left\|\log\left(1 + \frac{\frac{1}{n}\sum_{i=1}^n \varphi_i - \varphi}{\int \exp\left((f^* + g - C)/\varepsilon\right)d\beta(y)}\right)\right\|_{var}$$

$$= \left\|\log\left(1 + \frac{\frac{1}{n}\sum_{i=1}^n \varphi_i - \varphi}{\color{red}M(x)}\right)\right\|_{var},$$

where $\varphi_i(x) = \exp\left((f^*(x) + g(y_i) - C(x, y_i))/\varepsilon\right)$ and $\varphi(x) = \int \exp\left((f^*(x) + g(y) - C(x, y))/\varepsilon\right)d\beta(y)$. Note that for any $x$

$$\frac{\frac{1}{n}\sum_{i=1}^n \varphi_i(x) - \varphi(x)}{M(x)} = \frac{\exp\left(-T_{\hat{\beta}}(g)/\varepsilon\right)}{\exp\left(-T_\beta(g)/\varepsilon\right)} - 1 > -1.$$

Let $B(x) = \frac{\frac{1}{n}\sum_{i=1}^n \varphi_i(x) - \varphi(x)}{M(x)}$, then $\|\zeta_{\hat{\beta}}\|_{var} \leq 2\|\zeta_{\hat{\beta}}\|_\infty = 2\left\|\log(1 + B)\right\|_\infty$.

By Lemma 7 and the assumption that $n$ is lower bounded, $\frac{1}{n}\sum_{i=1}^n \varphi_i(x) - \varphi(x)$ is upper bounded with respect to $x$. Hence, there exists $-1 < L < U$, such that $L < B(x) < U$ for any $x$. Therefore, for any $x$

$$\log(1 + L) < \log(1 + B(x)) < \log(1 + U),$$

and $\|\zeta_{\hat{\beta}}\|_{var} < 2\max\{|\log(1 + L)|, |\log(1 + U)|\} := Q$ for all $t$.

On the event $\Omega_t = \{B(x) \leq \frac{1}{2}\}$, by applying $\log(1 + y) \leq -\log(1 - y) \leq 2y$ for $0 \leq y \leq \frac{1}{2}$

$$\|\zeta_{\hat{\beta}}\|_{var} \leq 2\max\left\{\log(1 + B(x)), -\log(1 - B(x))\right\} = -2\log(1 - B(x)) \leq 4B(x).$$

By the Markov inequality, $\mathbb{P}\left[B > \frac{1}{2}\right] \leq 2\mathbb{E}(B)$. Split the expectation $\mathbb{E}\left[\|\zeta_{\hat{\beta}}\|_{var}\right]$ into two parts, by Lemma 7 and the fact that $M$ is bounded from above and below with respect to $x$, there exists a constant $A_1(f^*, g^*, \varepsilon)$ depending on $f^*, g^*, \varepsilon$,

$$\mathbb{E}\|\zeta_{\hat{\beta}}\|_{var} = \mathbb{P}\left[B \leq \frac{1}{2}\right]\mathbb{E}\left[\|\zeta_{\hat{\beta}}\|_{var}\Big|B \leq \frac{1}{2}\right] + \mathbb{P}\left[B > \frac{1}{2}\right]\mathbb{E}\left[\|\zeta_{\hat{\beta}}\|_{var}\Big|B > \frac{1}{2}\right]$$

$$\leq \mathbb{E}(4B + 2BQ) = (4 + 2Q)\mathbb{E}B \tag{29}$$

$$\lesssim \frac{A_1(f^*, g^*, \varepsilon)}{\sqrt{n}}.$$

$\square$

The following lemma shows that the error in the variational norm at this step can be bounded using the error in the variational norm from the last step.

**Lemma 9.** *Given the empirical measures* $\hat{\alpha} = \frac{1}{n}\sum_{i=1}^n \delta_{x_i}$, $\hat{\beta} = \frac{1}{n}\sum_{i=1}^n \delta_{y_i}$, *where* $x_i \overset{iid}{\sim} \alpha$, $y_i \overset{iid}{\sim} \beta$ *and* $n \in \mathbb{Z}_+$. *Let* $f, g$ *be functions of* $C$-*transform, consider the update in the online Sinkhorn algorithm for* $\eta_t$ *at step* $t$,

$$\exp(-\hat{f}(x)/\varepsilon) = (1 - \eta_t)\exp(-f(x)/\varepsilon) + \eta_t\int\exp((g(y) - C(x, y))/\varepsilon)d\hat{\beta}(y), \tag{30}$$

$$\exp(-\hat{g}(x)/\varepsilon) = (1 - \eta_t)\exp(-g(x)/\varepsilon) + \eta_t\int\exp((\hat{f}(y) - C(x, y))/\varepsilon)d\hat{\alpha}(x). \tag{31}$$

Denote $U \overset{\text{def.}}{=} (f - f^*)/\varepsilon$, $V \overset{\text{def.}}{=} (g - g^*)/\varepsilon$, $\hat{U} \overset{\text{def.}}{=} (\hat{f} - f^*)/\varepsilon$, $\hat{V} \overset{\text{def.}}{=} (\hat{g} - g^*)/\varepsilon$, $\zeta_{\hat{\beta}} \overset{\text{def.}}{=} \left( T_\beta(g) - T_{\hat{\beta}}(g) \right)/\varepsilon$, and $\iota_{\hat{\alpha}} \overset{\text{def.}}{=} \left( T_\alpha(\hat{f}) - T_{\hat{\alpha}}(\hat{f}) \right)/\varepsilon$, then for $t$ large enough,

$$\|\hat{U}\|_{var} + \|\hat{V}\|_{var} \le (1 - \eta_t + \eta_t \kappa)(\|U\|_{var} + \|V\|_{var}) + \eta_t \left( \|\zeta_{\hat{\beta}}\|_{var} + \|\iota_{\hat{\alpha}}\|_{var} \right). \tag{32}$$

*Proof.* Multiply $\exp(f^*/\varepsilon)$ on both sides on equation 30 we get

$$\exp\left((-\hat{f} + f^*)/\varepsilon\right) = (1 - \eta_t)\exp\left((-f + f^*)/\varepsilon\right) + \eta_t \exp\left(f^*(x)/\varepsilon\right) \int \exp\left((g(y) - C(x,y))/\varepsilon\right) \mathrm{d}\hat{\beta}(y)$$

$$= (1 - \eta_t)\exp\left((-f + f^*)/\varepsilon\right) + \eta_t \exp\left(-T_{\hat{\beta}}(g)/\varepsilon + T_\beta(g^*)/\varepsilon\right)$$

$$= (1 - \eta_t)\exp\left((-f + f^*)/\varepsilon\right) + \eta_t \exp\left(\left(T_\beta(g) - T_\beta(g) - T_{\hat{\beta}}(g) + T_\beta(g^*)\right)/\varepsilon\right)$$

$$= (1 - \eta_t)\exp\left((-f + f^*)/\varepsilon\right) + \eta_t \exp\left(-T_\beta(g)/\varepsilon + T_\beta(g^*)/\varepsilon + \zeta_{\hat{\beta}}\right),$$

and similarly, multiply $\exp(g^*/\varepsilon)$ on both sides of equation 31,

$$\exp\left((-\hat{g} + g^*)/\varepsilon\right) = (1 - \eta_t)\exp\left((-g + g^*)/\varepsilon\right) + \eta_t \exp\left(-T_\alpha(\hat{f})/\varepsilon + T_\alpha(f^*)/\varepsilon + \iota_{\hat{\alpha}}\right) \tag{33}$$

where $(f^*, g^*)$ is the pair of optimal potentials.

Denote $\hat{U}^T \overset{\text{def.}}{=} \left( T_\alpha(\hat{f}) - T_\alpha(f^*) \right)/\varepsilon$ and $V^T \overset{\text{def.}}{=} \left( T_\beta(g) - T_\beta(g^*) \right)/\varepsilon$. Then, we can find an upper bound for $\max \hat{U}$,

$$\max \hat{U} = -\log \min \exp(-\hat{U})$$

$$= -\log\left(\min\left((1 - \eta_t)\exp(-U) + \eta_t \exp(-V^T + \zeta_{\hat{\beta}})\right)\right) \quad \text{by the update equation 30}$$

$$\le -\log\left((1 - \eta_t)\min \exp(-U) + \eta_t \min \exp(-V^T + \zeta_{\hat{\beta}})\right) \quad \text{by } \min f_1 + \min f_2 \le \min(f_1 + f_2)$$

$$\le -(1 - \eta_t)\log \min \exp(-U) - \eta_t \log \min \exp(-V^T + \zeta_{\hat{\beta}}) \quad \text{by Jensen Inequality}$$

$$= (1 - \eta_t)\max U + \eta_t \max\left(V^T - \zeta_{\hat{\beta}}\right),$$

and similarly,

$$\min \hat{U} \ge (1 - \eta_t)\min U + \eta_t \min\left(V^T - \zeta_{\hat{\beta}}\right),$$

$$\max \hat{V} \le (1 - \eta_t)\max V + \eta_t \max\left(\hat{U}^T - \iota_{\hat{\alpha}}\right),$$

$$\min \hat{V} \ge (1 - \eta_t)\min V + \eta_t \min\left(\hat{U}^T - \iota_{\hat{\alpha}}\right).$$

Therefore, by the contractivity of the soft-$C$ transform (Vialard, 2019, Proposition 19) that for the contractivity constant $\kappa = 1 - \exp(-L\mathrm{diam}(\mathcal{X})) < 1$ using Assumption 1, $\left\|V^T\right\|_{var} \le \kappa \|V\|_{var}$ and $\left\|\hat{U}^T\right\|_{var} \le \kappa \|\hat{U}\|_{var}$ for any $t$ (Mensch & Peyré, 2020, Lemma 1). Thus,

$$\|\hat{U}\|_{var} \le (1 - \eta_t)\|U\|_{var} + \eta_t \|V^T\|_{var} + \eta_t \|\zeta_{\hat{\beta}}\|_{var}$$

$$\le (1 - \eta_t)\|U\|_{var} + \eta_t \kappa \|V\|_{var} + \eta_t \|\zeta_{\hat{\beta}}\|_{var} \tag{34}$$

$$\|\hat{V}\|_{var} \le (1 - \eta_t)\|V\|_{var} + \eta_t \|\hat{U}^T\|_{var} + \eta_t \|\iota_{\hat{\alpha}}\|_{var}$$

$$\le (1 - \eta_t)\|V\|_{var} + \eta_t \kappa \|\hat{U}\|_{var} + \eta_t \|\iota_{\hat{\alpha}}\|_{var}. \tag{35}$$

Substitute equation 34 into the RHS of equation 35,

$$\|\hat{V}\|_{var} \le (1 - \eta_t)\|V\|_{var} + \eta_t \kappa \left((1 - \eta_t)\|U\|_{var} + \eta_t \kappa \|V\|_{var} + \eta_t \|\zeta_{\hat{\beta}}\|_{var}\right) + \eta_t \|\iota_{\hat{\alpha}}\|_{var}$$

$$= \left(1 - \eta_t + \eta_t^2 \kappa^2\right)\|V\|_{var} + (1 - \eta_t)\eta_t \kappa \|U\|_{var} + \eta_t^2 \kappa \|\zeta_{\hat{\beta}}\|_{var} + \eta_t \|\iota_{\hat{\alpha}}\|_{var}. \tag{36}$$

Add up equation 34 and equation 36

$$\|\hat{U}\|_{var} + \|\hat{V}\|_{var} \le (1 - \eta_t + \eta_t \kappa - \eta_t^2 \kappa)\|U\|_{var} + \left(1 - \eta_t + \eta_t \kappa + \eta_t^2 \kappa^2\right)\|V\|_{var}$$

$$+ \left(\eta_t + \eta_t^2 \kappa\right) \|\zeta_{\hat{\beta}}\|_{var} + \eta_t \|\iota_{\hat{\alpha}}\|_{var}.$$

For $t$ large enough,

$$\|\hat{U}\|_{var} + \|\hat{V}\|_{var} \leq \left(1 - \eta_t + \eta_t \kappa\right)\left(\|U\|_{var} + \|V\|_{var}\right) + \eta_t \left(\|\zeta_{\hat{\beta}}\|_{var} + \|\iota_{\hat{\alpha}}\|_{var}\right), \tag{37}$$

which holds for a possibly increased value of $\kappa$, as $\eta_t^2$ is negligible compared to $\eta_t$. □

Making use of a discrete version of Gronwall's lemma (Mischler, 2018, Lemma 5.1), we are able to show the upper bound of $a_t$ by the recursion relation $a_{t+1} \lesssim \left(1 - \eta_t + \eta_t \kappa\right) a_t + t^{\theta}$.

**Lemma 10.** *Given a sequence $a_n$ such that $a_{t+1} \lesssim \left(1 - \eta_t + \eta_t \kappa\right) a_t + t^{\theta}$. Then*

$$a_{t+1} \lesssim (a_1 + C_1) \exp\left(C_2 t^{b+1}\right) + t^{\theta - b}, \tag{38}$$

*where $C_1 = \frac{1}{-\theta - 1} > 0$ and $C_2 = \frac{\kappa - 1}{b+1} < 0$*

*Proof.* Summing over $t$, by the discrete version of Gronwall lemma (Mischler, 2018, Lemma 5.1), we have

$$a_{t+1} \lesssim \prod_{i=1}^{t} \left(1 - \eta_i + \eta_i \kappa\right) a_1 + \sum_{i=1}^{t-1} \left(\prod_{j=i+1}^{t} \left(1 - \eta_j + \eta_j \kappa\right)\right) i^{\theta} + t^{\theta}$$

$$\overset{\text{def.}}{=} A_{1,t} a_1 + A_{2,t} + t^{\theta},$$

where

$$A_{1,t} = \prod_{i=1}^{t} \left(1 - \eta_i + \eta_i \kappa\right),$$

$$A_{2,t} = \sum_{i=1}^{t-1} \left(\prod_{j=i+1}^{t} \left(1 - \eta_j + \eta_j \kappa\right)\right) i^{\theta}.$$

First consider the term $A_{1,t}$, and taking the logarithm on it, using $\log(1 + x) < x$,

$$\log A_{1,t} = \sum_{i=1}^{t} \log\left(1 + (\kappa - 1)\eta_i\right) \leq \sum_{i=1}^{t} (\kappa - 1)\eta_i$$

$$< (\kappa - 1)\int_1^{t+1} x^b \mathrm{d}x = \frac{\kappa - 1}{b+1}\left((t+1)^{b+1} - 1\right). \tag{39}$$

Thus, $A_{1,t} \leq \exp\left(\frac{\kappa - 1}{b+1}\left((t+1)^{b+1} - 1\right)\right)$.

Now, take a look at the term $A_{2,t}$. Following the proof of (Moulines & Bach, 2011, Theorem 1), for any $1 < m < t - 1$,

$$A_{2,t} = \sum_{i=1}^{m} \left(\prod_{j=i+1}^{t} \left(1 - \eta_j + \eta_j \kappa\right)\right) i^{\theta} + \sum_{i=m+1}^{t-1} \left(\prod_{j=i+1}^{t} \left(1 - \eta_j + \eta_j \kappa\right)\right) i^{\theta}. \tag{40}$$

The first term $\sum_{i=1}^{m} \left(\prod_{j=i+1}^{t} \left(1 - \eta_j + \eta_j \kappa\right)\right) i^{\theta} \leq \exp\left(\sum_{j=m+1}^{t} (\kappa - 1)\eta_j\right) \sum_{i=1}^{m} i^{\theta}$, and the second term

$$\sum_{i=m+1}^{t-1} \left(\prod_{j=i+1}^{t} \left(1 - \eta_j + \eta_j \kappa\right)\right) i^{\theta} \leq m^{\theta - b} \sum_{i=m+1}^{t-1} \prod_{j=i+1}^{t} \left(1 - (1 - \kappa)\eta_j\right)\eta_i$$

$$= \frac{m^{\theta - b}}{1 - \kappa} \sum_{i=m+1}^{t-1} \left[\prod_{j=i+1}^{t} \left(1 - (1-\kappa)\eta_j\right) - \prod_{j=i}^{t} \left(1 - (1-\kappa)\eta_j\right)\right] \tag{41}$$

$$\leq \frac{m^{\theta - b}}{1 - \kappa}\left[1 - \prod_{j=m+1}^{t-1} \left(1 - (1-\kappa)\eta_j\right)\right] < \frac{m^{\theta - b}}{1 - \kappa}.$$

Therefore,

$$A_{2,t} < \exp\left((\kappa-1)\int_{m+1}^{t+1} x^b \mathrm{d}x\right)\int_1^m x^\theta \mathrm{d}x + \frac{m^{\theta-b}}{1-\kappa}$$

$$= \exp\left(\frac{\kappa-1}{b+1}\left((t+1)^{b+1}-(m+1)^{b+1}\right)\right)\frac{m^{\theta+1}-1}{\theta+1} + \frac{m^{\theta-b}}{1-\kappa}.$$

Take $m = \frac{t}{2}$,

$$A_{2,t} \lesssim \exp\left(\frac{\kappa-1}{b+1}\left((t+1)^{b+1}-(t/2)^{b+1}\right)\right)\frac{1}{-\theta-1} + \frac{t^{\theta-b}}{1-\kappa}. \tag{42}$$

Combine Equations equation 39 and equation 42

$$a_{t+1} \lesssim \left(a_1 + \frac{1}{-\theta-1}\right)\exp\left(\frac{\kappa-1}{b+1}t^{b+1}\right) + t^{\theta-b}. \tag{43}$$

$\square$

## A.3 PROOF OF THEOREM 1

*Proof.* Let $U_t \overset{\text{def.}}{=} \left(f_t - f^*\right)/\varepsilon$, $V_t \overset{\text{def.}}{=} \left(g_t - g^*\right)/\varepsilon$, $e_t \overset{\text{def.}}{=} \|U_t\|_{var} + \|V_t\|_{var}$, $\zeta_{\hat{\beta}_t} \overset{\text{def.}}{=} \left(T_\beta(g_t) - T_{\hat{\beta}_t}(g_t)\right)/\varepsilon$ and $\iota_{\hat{\alpha}_t} \overset{\text{def.}}{=} \left(T_\alpha(f_{t+1}) - T_{\hat{\alpha}_t}(f_{t+1})\right)/\varepsilon$. By Lemma 9,

$$e_{t+1} \le \left(1 - \eta_t + \eta_t \kappa\right)e_t + \eta_t\left(\|\zeta_{\beta_t}\|_{var} + \|\iota_{\alpha_t}\|_{var}\right). \tag{44}$$

Define $M_t = \int \exp((f^* + g_t - c)/\varepsilon)\mathrm{d}\beta$, then

$$M_t = \int \exp((f^* + g^* - g^* + g_t - c)/\varepsilon)\mathrm{d}\beta$$

$$= \int \exp((-g^* + g_t)/\varepsilon)\mathrm{d}\beta,$$

and by Lemma 6, there exists $\delta = \sup_{x \in \mathcal{X}}|f_{t_0} - f^*| > 0$ for $t_0 < t$, such that $\exp(-\delta/\varepsilon) < M_t < \exp(\delta/\varepsilon)$. Apply Lemma 8,

$$\mathbb{E}\left\|\zeta_{\hat{\beta}_t}\right\|_{var} \lesssim \frac{A_1(f^*, g^*, \varepsilon)}{\sqrt{b_t}},$$

$$\mathbb{E}\left\|\iota_{\hat{\alpha}_t}\right\|_{var} \lesssim \frac{A_2(f^*, g^*, \varepsilon)}{\sqrt{b_t}}, \tag{45}$$

where $A_1(f^*, g^*, \varepsilon), A_2(f^*, g^*, \varepsilon)$ are constants depending on $f^*, g^*, \varepsilon$.

Denote $S = A_1(f^*, g^*, \varepsilon) + A_2(f^*, g^*, \varepsilon)$, then taking expectation on equation 44 with Equations equation 45

$$\mathbb{E}e_{t+1} \le \left(1 - \eta_t + \eta_t \kappa\right)\mathbb{E}e_t + \eta_t\mathbb{E}\left(\|\zeta_{\beta_t}\|_{var} + \|\iota_{\alpha_t}\|_{var}\right)$$

$$\lesssim \left(1 - \eta_t + \eta_t \kappa\right)\mathbb{E}e_t + \frac{S\eta_t}{\sqrt{b_t}}. \tag{46}$$

By Lemma 10, we have

$$\mathbb{E}e_{t+1} \lesssim \left(\mathbb{E}e_1 + \frac{S}{a-b-1}\right)\exp\left(\frac{\kappa-1}{b+1}t^{b+1}\right) + t^{-a}, \tag{47}$$

whose RHS converges to 0 as $t \to \infty$

Consider the upper bound equation 47 for $\delta_{t+1}$,

$$\delta_{t+1} \lesssim \left(\delta_1 + \frac{S}{a-b-1}\right)\exp\left(\frac{\kappa-1}{b+1}t^{b+1}\right) + t^{-a}.$$

Note that the total sample size at step $t$ is $n_t = \sum_{i=1}^{t} i^{2a} = O(t^{2a+1})$, thus we can rewrite $t$ in terms of $n_t$, that is $t = n_t^{\frac{1}{2a+1}}$. The first term decays rapidly with the rate $O\left(\exp\left(-n^{\frac{b+1}{2a+1}}\right)\right)$, since $\frac{\kappa-1}{b+1} < 0$. Taking $t(N) = \lfloor N^{\frac{1}{2a+1}} \rfloor$, then when $t^{\frac{b+1}{2a+1}} \gg 1$,

$$\delta_N \lesssim \left(\delta_1 + \frac{S}{a-b-1}\right)\exp\left(\frac{\kappa-1}{b+1}N^{\frac{b+1}{2a+1}}\right) + N^{-\frac{a}{2a+1}} = O\left(N^{-\frac{1}{2+1/a}}\right),$$

which is bounded by $O\left(N^{-1/2}\right)$. $\qquad\qquad\square$

### A.4 Proof of Theorem 3

*Proof.* This proof follows up on the proof of Theorem 1.

Recall the following terms regarding $f_t$ and $g_t$,

$$U_t \overset{\text{def.}}{=} (f_t - f^*)/\varepsilon, \qquad\qquad V_t \overset{\text{def.}}{=} (g_t - g^*)/\varepsilon,$$
$$U_t^T \overset{\text{def.}}{=} (T_\alpha(f_t) - T_\alpha(f^*))/\varepsilon, \qquad\qquad V_t^T \overset{\text{def.}}{=} (T_\beta(g_t) - T_\beta(g^*))/\varepsilon,$$

and further define the corresponding terms regarding $\hat{f}_t$ and $\hat{g}_t$,

$$\hat{U}_t \overset{\text{def.}}{=} (\hat{f}_t - f^*)/\varepsilon, \qquad\qquad \hat{V}_t \overset{\text{def.}}{=} (\hat{g}_t - g^*)/\varepsilon,$$
$$\hat{U}_t^T \overset{\text{def.}}{=} (T_\alpha(\hat{f}_t) - T_\alpha(f^*))/\varepsilon, \qquad\qquad \hat{V}_t^T \overset{\text{def.}}{=} (T_\beta(\hat{g}_t) - T_\beta(g^*))/\varepsilon,$$
$$\hat{\zeta}_{\hat{\beta}_t} \overset{\text{def.}}{=} \left(T_\beta(\hat{g}_t) - T_{\hat{\beta}_t}(\hat{g}_t)\right)/\varepsilon, \qquad\qquad \hat{\iota}_{\hat{\alpha}_t} \overset{\text{def.}}{=} \left(T_\alpha(f_{t+1}) - T_{\hat{\alpha}_t}(f_{t+1})\right)/\varepsilon.$$

Notice that under Assumption 4, $\left\|(f_t - \hat{f}_t)/\varepsilon\right\|_{var} = O\left(t^{-a+b}\right)$. Suppose that

$$\max\left\{\sup_x |\hat{f}_t - f^*|, \sup_y |\hat{g}_t - g^*|\right\} < \delta,$$

then by Lemma 6, $\sup_x |f_{t+1} - f^*| < \delta$. Thus,

$$\begin{aligned}
\sup_x |\hat{f}_{t+1} - f^*| &= \sup_x |\hat{f}_{t+1} - f_{t+1} + f_{t+1} - f^*| \\
&\le \sup_x |f_{t+1} - f^*| + \sup_x |\hat{f}_{t+1} - f_{t+1}| \\
&\lesssim \sup_x |f_{t+1} - f^*| + t^{-a+b} < \delta + t^{-a+b}.
\end{aligned}$$

Let $\delta = \max\left\{\sup_x |\hat{f}_1 - f^*|, \sup_y |\hat{g}_1 - g^*|\right\}$, we have

$$\sup_x |\hat{f}_{t+1} - f^*| \le \delta + \sum_{i=1}^{t} i^{-a+b},$$

which is bounded from above and below as $-a+b+1 < 0$.

Notice that $\hat{g}_t$ and $f_{t+1}$ are $\varepsilon L$-Lipschitz, and apply Lemma 8,

$$\mathbb{E}\left\|\zeta_{\hat{\beta}_t}\right\|_{var} \lesssim \frac{A_1(f^*, g^*, \varepsilon)}{\sqrt{b_t}},$$
$$\mathbb{E}\left\|\iota_{\hat{\alpha}_t}\right\|_{var} \lesssim \frac{A_2(f^*, g^*, \varepsilon)}{\sqrt{b_t}},$$

where $A_1(f^*, g^*, \varepsilon), A_2(f^*, g^*, \varepsilon)$ are constants depending on $f^*, g^*, \varepsilon$.

We have the following relation

$$\hat{U}_{t+1} = (\hat{f}_{t+1} - f_{t+1})/\varepsilon + (f_{t+1} - f^*)/\varepsilon \overset{\text{def.}}{=} \text{err}_{f_{t+1}} + u_{t+1},$$

$$\hat{V}_{t+1} = \left(\hat{g}_{t+1} - g_{t+1}\right)/\varepsilon + \left(g_{t+1} - g^*\right)/\varepsilon \stackrel{\text{def.}}{=} \mathrm{err}_{g_{t+1}} + V_{t+1},$$

where $\mathrm{err}_{f_{t+1}} = \left(\hat{f}_{t+1} - f_{t+1}\right)/\varepsilon$ and $\mathrm{err}_{g_{t+1}} = \left(\hat{g}_{t+1} - g_{t+1}\right)/\varepsilon$, and

$$\left\|\hat{U}_{t+1}\right\|_{var} \leqslant \left\|\mathrm{err}_{f_{t+1}}\right\|_{var} + \|U_{t+1}\|_{var}, \qquad (48)$$

$$\left\|\hat{V}_{t+1}\right\|_{var} \leqslant \left\|\mathrm{err}_{g_{t+1}}\right\|_{var} + \|V_{t+1}\|_{var}. \qquad (49)$$

Recall from Theorem 1 that $e_t \stackrel{\text{def.}}{=} \|U_t\|_{var} + \|V_t\|_{var}$, and define $\hat{e}_t \stackrel{\text{def.}}{=} \|\hat{U}_t\|_{var} + \|\hat{V}_t\|_{var}$. By Lemma 9, for $t$ large enough

$$e_{t+1} \leqslant \left(1 - \eta_t + \eta_t \kappa\right) \hat{e}_t + \eta_t \left(\|\hat{\zeta}_{\hat{\beta}_t}\|_{var} + \|\hat{\iota}_{\hat{\alpha}_t}\|_{var}\right). \qquad (50)$$

Thus, by the inequalities equation 48 and equation 50,

$$\begin{aligned}
\hat{e}_{t+1} &\leqslant e_{t+1} + \left\|\mathrm{err}_{f_{t+1}}\right\|_{var} + \left\|\mathrm{err}_{g_{t+1}}\right\|_{var} \\
&\leqslant \left(1 - \eta_t + \eta_t \kappa\right) \hat{e}_t + \eta_t \left(\|\hat{\zeta}_{\hat{\beta}_t}\|_{var} + \|\hat{\iota}_{\hat{\alpha}_t}\|_{var}\right) + \left\|\mathrm{err}_{f_{t+1}}\right\|_{var} + \left\|\mathrm{err}_{g_{t+1}}\right\|_{var}
\end{aligned} \qquad (51)$$

Take expectations on both sides of equation 51, we have

$$\begin{aligned}
\mathbb{E}\hat{e}_{t+1} &\leqslant \mathbb{E}e_{t+1} + \left\|\mathrm{err}_{f_{t+1}}\right\|_{var} + \mathbb{E}\left\|\mathrm{err}_{g_{t+1}}\right\|_{var} \\
&\lesssim \left(1 - \eta_t + \eta_t \kappa\right) \mathbb{E}\hat{e}_t + \frac{S\eta_t}{\sqrt{b_t}} + \mathbb{E}\left\|\mathrm{err}_{f_{t+1}}\right\|_{var} + \mathbb{E}\left\|\mathrm{err}_{g_{t+1}}\right\|_{var},
\end{aligned}$$

where $S = A_1(f^*, g^*, \varepsilon) + A_2(f^*, g^*, \varepsilon)$.

Notice that $\left\|\mathrm{err}_{f_{t+1}}\right\|_{var} = O\left(t^{-a+b}\right)$ under Assumption 4. By Lemma 10 with $\theta = b - a$,

$$\mathbb{E}\hat{e}_{t+1} \lesssim \left(\mathbb{E}e_1 + \frac{1}{-\theta - 1}\right) \exp\left(\frac{\kappa - 1}{b + 1} t^{b+1}\right) + \frac{1}{(1 - \kappa)} t^{-a}, \qquad (52)$$

where the right-hand side goes to 0 as $t \to \infty$. $\qquad\qquad \square$

## A.5 COMPRESSION ERRORS

### A.5.1 GAUSS QUADRATURE (GQ)

A quadrature rule uses a sum of specific points with assigned weights as an approximation to an integral, which are optimal with respect to a certain polynomial degree of exactness Gautschi (2004). The $m$-point Gauss quadrature rule for $\mu$ can be expressed as

$$\int_{\mathbb{R}} f(y)\, d\mu(y) = \sum_{i=1}^{m} w_i f\left(\hat{y}_i\right) + R_m\left(f\right),$$

for weights $w_i$ and points $\hat{y}_i$, where the remainder term $R_m$ satisifes $R_m\left(\mathbb{P}_{2m-1}\right) = 0$, and therefore the sum approximation on the RHS matches the integral value on the LHS for $f \in \mathbb{P}_{2m-1}$.

In general, by (Gautschi, 2004, Corollary to Theorem 1.48), the error term $R_m$ can be expressed as

$$R_m(f) = M\frac{f^{(2m)}(\xi)}{(2m)!}, \quad \text{some } \xi \in \mathbb{R}$$

where $M = \int_{\mathbb{R}} \left[\pi_n\left(t; d\mu\right)\right]^2 d\mu(t)$ and $\pi_n\left(\cdot; d\mu\right)$ is the numerator polynomial (Gautschi, 2004, Definition 1.35).

In our case, $f = K_x/v_t$. Note that $K_x$ and $v_t$ are smooth (both have the same regularity as $e^{-C(x,y)/\varepsilon}$; see equation 8). Moreover, note that $v_t$ is uniformly bounded away from 0 since $g_t$ is uniformly bounded. For $K_x(\xi) = \exp\left(-\frac{C(x,\xi)}{\varepsilon}\right)$ and, by the closed form of Gaussian functions,

$$K_x^{(2m)}(\xi) = \left(\frac{1}{\varepsilon}\right)^m \exp\left(-\frac{(x-\xi)^2}{\varepsilon}\right) H_{2m}\left(\frac{x-\xi}{\sqrt{\varepsilon}}\right) = O\left(\frac{(2m)!}{m!\varepsilon^m}\right),$$

where $H_n$ is the Hermite polynomial of $n$th order. It follows, by the Leibniz rule and the Faa di Bruno formula that, $R_m(f) = O\left(\frac{1}{\varepsilon^m m!}\right)$. Hence,

$$u_t(x) - \hat{u}_t(x) = \int \frac{K_x}{v_t}(y)\mathrm{d}\mu(y) - \int \frac{K_x}{v_t}(y)\mathrm{d}\hat{\mu}(y) = O\left(\frac{1}{\varepsilon^m m!}\right).$$

Note that $f_t(x) = -\varepsilon \log(u_t(x))$. By the mean-value theorem, $|\log a - \log b| \le \frac{1}{a}|a - b|$ for $0 < a < b$. Further, $u_t$ is bounded away from zero (as a continuous and positive function on a compact set). Hence, we may find a Lipschitz constant $L'$ such that $|f_t(x) - \hat{f}_t(x)| \le L'|u_t(x) - \hat{u}_t(x)|$ for all $x \in \mathcal{X}$. Hence, Assumption 4 holds for any $\zeta > 0$.

### A.5.2 FOURIER METHOD

Consider $C(x, y) = \|x - y\|^2$. Take Fourier moments of $\int \frac{K_x}{v_t}(y)\mathrm{d}\mu(y)$,

$$\int \exp(-ikx) \int \frac{K_x}{v_t}(y)\,d\mu(y)\,dx = \int \frac{\hat{K}_y(k)}{v_t(y)}\mathrm{d}\mu(y) = \sqrt{\varepsilon\pi}\exp\left(-\frac{\varepsilon k^2}{4}\right)\int \frac{\exp(-iky)}{v_t(y)}\mathrm{d}\mu(y), \quad (53)$$

as $\hat{K}_y(k)$ for $K_x(y) = \exp\left(-\frac{(x-y)^2}{\varepsilon}\right)$ is given by

$$\begin{aligned}
\hat{K}_y(k) &= \int \exp(-ikx)\exp\left(-\frac{(x-y)^2}{\varepsilon}\right)dx \\
&= \exp(-iky)\int \exp(-ikz)\exp\left(-\frac{z^2}{\varepsilon}\right)dz \\
&= \exp(-iky)\sqrt{\varepsilon\pi}\exp\left(-\frac{\varepsilon k^2}{4}\right).
\end{aligned} \quad (54)$$

By the Fourier inversion theorem,

$$u_t(x) = \exp(-f_t(x)/\varepsilon) = \int \exp(ikx)\sqrt{\varepsilon\pi}\exp\left(-\frac{\varepsilon k^2}{4}\right)\int \frac{\exp(-iky)}{v_t(y)}\mathrm{d}\mu(y)\,\mathrm{d}k. \quad (55)$$

The compression error becomes

$$u_t(x) - \hat{u}_t(x) = \int \frac{K_x}{v_t}(y)d\mu(y) - \int \frac{K_x}{v_t}(y)d\hat{\mu}(y) = \int \frac{\varphi_x(z)}{v_t(y)}\mathrm{d}z, \quad (56)$$

where

$$\varphi_x(k) = \exp(izk)\sqrt{\varepsilon\pi}\exp\left(-\frac{\varepsilon k^2}{4}\right)\int \exp(-iky)\,\mathrm{d}(\mu - \hat{\mu})(y), \quad (57)$$

and note that, by construction, $\varphi_x(k) = 0$ for all $k \in \Omega$. The problem of finding the compression thus becomes a problem of evaluating the integral $\int \varphi_x(z)\,\mathrm{d}z$. Similarly to Gauss quadrature, we want to find $k \in \Omega = \{k_1, \cdots, k_m\}$ such that the $\frac{1}{m}\sum_{i=1}^{m}\varphi_x(k_i)$ is an approximation to $\int \varphi_x(z)\,\mathrm{d}z$.

Let $\Omega = \{k_1, \cdots, k_m\}$ be the set of $n$ elements that are QMC sampled from $X \sim \mathcal{N}\left(0, \frac{2}{\varepsilon}I\right)$. In practice, we use the implementation of `SciPy` (Virtanen et al., 2020). Define $\chi(z) = \exp\left(\frac{\varepsilon z^2}{4}\right)\varphi_x(z)$. Notice that

$$\begin{aligned}
\mathbb{E}(\chi(X)) &= \int_{\mathbb{R}^d}(2\pi)^{-d/2}(2/\varepsilon)^{-d/2}\exp\left(-\frac{\varepsilon}{4}z^2\right)\psi(z)\,\mathrm{d}z \\
&= \left(\frac{4\pi}{\varepsilon}\right)^{-d/2}\int_{\mathbb{R}^d}\varphi_x(z)\,\mathrm{d}z,
\end{aligned} \quad (58)$$

and thus,

$$\int_{\mathbb{R}^d}\varphi_x(z)\,\mathrm{d}z = \left(\frac{4\pi}{\varepsilon}\right)^{d/2}\mathbb{E}\chi(X). \quad (59)$$

Following (Kuo & Nuyens, 2016, Section 4.1), let $\Psi(x)$ denote the cumulative distribution functions for $\mathcal{N}(0,1)$. Then $\Psi(\sqrt{\frac{\varepsilon}{2}}X_i) \sim \mathcal{U}(0,1)$ and

$$\mathbb{E}\left(\chi(X)\right) = \int_{[0,1]^d} \chi\left(A\Psi^{-1}(z)\right)dz, \tag{60}$$

where $A = \sqrt{\frac{2}{\varepsilon}}I$, $\Psi^{-1}(z) = \left(\Psi^{-1}(z^1), \cdots, \Psi^{-1}(z^d)\right)$. Denote $h(z) = \chi\left(A\Psi^{-1}(z)\right)$, and $I_d(h) = \int_{[0,1]^d} h(z)\,dt$, where $d$ is the dimension, and $Q_{m,d}(h) = \frac{1}{m}\sum_{i=1}^{m} h(k_i)$. As $y \in \mathcal{X}$, a compact domain, the first derivative of $\chi(k)$ is bounded and the integrand $h$ has bounded variation. Hence, by (Kuo & Nuyens, 2016, Section 1.3),

$$\left|I_d(h) - Q_{m,d}(h)\right| = O\left(\frac{|\log m|^d}{m}\right). \tag{61}$$

Given that for $k \in \Omega$, $\varphi_x(k) = 0$ and hence, $Q_{m,d}(h) = 0$. Then, we know

$$\int_{\mathbb{R}^d} \varphi_x(z)\,dz = \left(\frac{4\pi}{\varepsilon}\right)^{d/2} I_d(h) \leqslant \left(\frac{4\pi}{\varepsilon}\right)^{d/2}\left(Q_{m,d}(h) + \left|I_d(h) - Q_{m,d}(h)\right|\right)$$
$$= O\left(\left(\frac{4\pi}{\varepsilon}\right)^{d/2}\frac{|\log m|^d}{m}\right). \tag{62}$$

Choosing $k$ in equation 13 via Gaussian QMC, the compression error is then

$$u_t(x) - \hat{u}_t(x) = \exp\left(-f_t(x)/\varepsilon\right) - \exp\left(-\hat{f}_t(x)/\varepsilon\right) = O\left(\frac{|\log m|^d}{m}\right). \tag{63}$$

If we take $\zeta < 1$, then $\frac{|\log m|^d}{m} = o\left(m^{-\zeta}\right)$. Again applying the Lipschitz argument, $f_t(x) - \hat{f}_t(x) = O\left(\frac{|\log m|^d}{m}\right)$ and Assumption 4 holds.

### A.6  PROOF OF PROPOSITION 4

*Proof.* **Cost of online Sinkhorn** Now we calculate the complexity of Algorithm 2 up to step $T$. Recalled that in Section 3, the computational complexity of Algorithm 1 is $\mathscr{C} = O\left(dT^{4a+2}\right)$. By Theorem 1, the error at step $T$ is $\delta = O(T^{-a})$ (where the hidden constant may depend on dimension). Taking $T = O(\delta^{-1/a})$, we have $\mathscr{C} = O\left(\delta^{-\left(4+\frac{2}{a}\right)}\right)$.

**Cost of compressed online Sinkhorn**

Assuming $m_t = t^{\frac{a-b}{\zeta}}$, the total computational cost of Algorithm 2 is

$$\hat{\mathscr{C}} = \sum_{t=1}^{T}\left(C(n_{t-1}, m_t) + db_t m_t\right)$$
$$= \sum_{t=1}^{T}\left(m_t^3 + d(m_{t-1} + b_{t-1})m_t + db_t m_t\right)$$
$$= O\left(\sum_{t=1}^{T}\left(t^{\frac{3(a-b)}{\zeta}} + d(t-1)^{2a}t^{\frac{a-b}{\zeta}} + dt^{2a+\frac{a-b}{\zeta}}\right)\right)$$
$$= \begin{cases} O\left(dT^{2a+\frac{a-b}{\zeta}+1}\right), & \text{for } \zeta \geqslant \frac{a-b}{a} \\ O\left(T^{\frac{3(a-b)}{\zeta}+1}\right), & \text{for } \zeta < \frac{a-b}{a} \end{cases}.$$

Under Assumption 4, the error at step $T$ is $\delta = O(T^{-a})$. Taking $T = \delta^{-1/a}$, we have

$$\hat{\mathscr{C}} = \begin{cases} O\left(\delta^{-\left(2+\frac{a-b}{a\zeta}+\frac{1}{a}\right)}\right), & \text{for } \zeta \geqslant \frac{a-b}{a}, \\ O\left(\delta^{-\left(\frac{3(a-b)}{a\zeta}+\frac{1}{a}\right)}\right), & \text{for } \zeta < \frac{a-b}{a}. \end{cases}$$

The big-$O$ constant for the accuracy $\delta$ may depend on dimension. □

Notice that the ratio of $\hat{\mathscr{C}}$ and $\mathscr{C}$,

$$\frac{\hat{\mathscr{C}}}{\mathscr{C}} = \begin{cases} O\left(\delta^{2+\frac{1}{a}-\frac{a-b}{a\zeta}}\right), & \text{for } \zeta \geq \frac{a-b}{a}, \\ O\left(\delta^{4+\frac{1}{a}-\frac{3(a-b)}{a\zeta}}\right), & \text{for } \zeta < \frac{a-b}{a}, \end{cases} \tag{64}$$

and the exponent of the ratio is positive when $\zeta > \frac{3(a-b)}{4a+1}$, which means the asymptotical convergence of the Compressed Online Sinkhorn is improved compared to the Online Sinkhorn.

