# OpenReview forum: "Compressed Online Sinkhorn"
_ICLR.cc/2024/Conference — Submitted to ICLR 2024_

### Official Review · Reviewer_XcQQ · 2023-10-31

**Soundness:** 2 fair
**Presentation:** 2 fair
**Contribution:** 2 fair
**Rating:** 5
**Confidence:** 3

**Summary:**

Sinkhorn is a popular algorithm for calculating entropic-regularised OT distances. However, before applying the Sinkhorn algorithm to the discrete probability measures, one should first draw a large stream of data from the probability distributions. This situation has been improved by online Sinkhorn method, which continuously samples data from two probability distributions in batches and iteratively computes the results. This paper revisits the recently introduced online Sinkhorn algorithm of $Mensch\ \& \ Peyre \ (2020)$ and rises two improvements.

1. This work presents a new convergence rate for the the online Sinkhorn algorithm, which is faster than the previous rate under certain parameter choices.

2. Under two new assumptions, the authors propose the compressed online Sinkhorn algorithm which combines measure compression techniques with the online Sinkhorn algorithm. Under certain parameter values, the new algorithm theoretically has a faster speed and smaller error than the previous online Sinkhorn algorithm.

The authors also provide experimental results to show the numerical gains of these two improvements.

**Strengths:**

1. The authors provide clear theoretical analysis for the issues present in  Mensch and Peyre  (2020)  and their new method.
2. The presentation of the results are clear.

**Weaknesses:**

1. The authors do not discuss the performance of the algorithm in high-dimensional situations. Real-world data often has a high dimensionality (such as datasets of images and amino acid sequences), but the authors do not discuss cases where $d > 5$. In \textbf{section A.5.2}, for data of dimension $d$, the compression error is $O(\frac{|\log m|^{d}}{m})$, which may too large in high-dimensional situations (i.e., assumption 4 with a large coefficient for $O(m_t^{-\zeta})$). In fact, in the experimental part,  Figure 2(c), when d=5, the online Sinkhorn algorithm already has lower error than the new compressed online Sinkhorn algorithm.

2. The Algorithm 2 proposed by the author improves the speed compared to the original online Sinkhorn algorithm by using measure compression technique to compress $u_t$ and $v_t$ from $n$ atoms to $m$. However, there is a trade-off between accuracy and speed. According to assumption 4, the smaller the value of $m$, the faster the algorithm but the larger the error. The article seems to lack a detailed discussion on this matter, such as how to choose an appropriate batch size $m_t$ when solving actual OT problems.

3. The experimental sections lack the application of the algorithm on real-world data and more complex distributions.

**Questions:**

The new convergence rate of the proposed online Sinkhorn algorithm in this paper is better than the original rate when $a > -b$. Additionally, Algorithm 2 proposed in this paper is theoretically more efficient than Algorithm 1 when $\zeta > \frac{3(a-b)}{4a+1}$. However, the paper lacks an explanation on how to choose specific values for parameters $a$, $b$, and $\zeta$, which makes the experiments in this paper somewhat less persuasive. Please explain why specific parameter values are chosen in the experiments.

---

> ### Author Response · Authors · 2023-11-22
>
> We appreciate your thorough review of our paper, and your feedback has been very valuable to us. We've  addressed each of your points:
>
> > The authors do not discuss the performance of the algorithm in high-dimensional situations.
>
> As mentioned above, we have clarified the issues with dimension when applying  QMC-based Fourier method and explained the dimension dependence of the error bound. Asymptotically, the compressed online Sinkhorn can be more efficient. It is more difficult  however to get into the asymptotic regime with the dimension $d$ is high.
>
> > However, there is a trade-off between accuracy and speed. According to assumption 4, the smaller the value of $m$, the faster the algorithm but the larger the error. The article seems to lack a detailed discussion on this matter, such as how to choose an appropriate batch size $m_t$ when solving actual OT problems.
>
> We made some changes to Assumption 4, please check the new version. The choice of $m_t$ depends on the compression algorithm, such that Assumption 4 holds. The purpose of this assumption is that the compression error does not exceed the online Sinkhorn error in Algorithm 1, and we focus on the overall efficiency of the compressed algorithm. This allows for an increased error if there is sufficient reduction in computational effort to allow more iterations. This is now shown more effectively in the numerical experiments, where we ensure the error in the objective for the different algorithms match in the Figure 2.
>
>
> > The experimental sections lack the application of the algorithm on real-world data and more complex distributions.
>
> Thanks for this valid point. This work is a first attempt at a compression method and the improvements are mainly in the asymptotic regime and lower dimensional setting. As future work, we plan on refining this by investigating adaptive choices of the parameters to enhance the fast transient behaviour in the error bound, making the algorithm more competitive for real-world data.

---

> > ### Comment · Reviewer_XcQQ · 2023-11-23
> > **Thanks**
> >
> > Thanks for your response, and I will consider it in the next discussion stage.

---

### Official Review · Reviewer_q5md · 2023-11-01

**Soundness:** 4 excellent
**Presentation:** 3 good
**Contribution:** 3 good
**Rating:** 6
**Confidence:** 3

**Summary:**

This paper adds a compression step on top of the online sinkhorn algorithm of Mencsh and Peyré in regimes in which some measure compression can be perform. They show two such compression schemes: gaussian quadrature and to Fourier moments compression. They analyze the method theoretically and provide numerical evidence of its lower runtime while the observed error empirically is comparable to the one of the uncompressed method.

**Strengths:**

The authors provide two settings for which their compression can be implemented: Gaussian quadrature and Fourier moments compression.

They fix a minor error in a proof of a previous paper on online Sinkhorn.

Numerical evidence is presented.

The paper is well written.

**Weaknesses:**

The experiments are done in settings of very low dimensionality. For the one of greater dimension (d=5), the uncompressed method starts to look quite better.

**Questions:**

I wonder if what I mentioned above regarding the uncompressed method working better and better with increasing dimension is a general trend.

---

> ### Author Response · Authors · 2023-11-22
>
> Thank you for your thorough review of our paper. Your feedback has been invaluable to us. We've  addressed your points:
>
> > The experiments are done in settings of very low dimensionality. For the one of greater dimension (d=5), the uncompressed method starts to look quite better. I wonder if what I mentioned above regarding the uncompressed method working better and better with increasing dimension is a general trend.
>
> We have clarified the issues with dimension when applying  QMC method. Asymptotically, the compressed online Sinkhorn can be more efficient. It is difficult however to get into the asymptotic regime due to the small contraction rate ($\kappa\approx 1$) and available compression methods.

---

### Official Review · Reviewer_5NvN · 2023-11-02

**Soundness:** 3 good
**Presentation:** 3 good
**Contribution:** 3 good
**Rating:** 6
**Confidence:** 3

**Summary:**

The Sinkhorn algorithm for entropy-regularized optimal transport is well-known, but very computationally complex. The present paper considers two online variants of this method. The first one comes from a Mensch and Peyré: the bound here is sometimes worse, sometimes better, but the present paper claims (convincingly) through theory and simulations that the previous bound was wrong. The second variant is a compressed version of online Sinkhorn where the random samples are compressed eg. via quadrature techniques.

**Strengths:**

The idea of compressing measures seems very interesting from an algorithmic point of view. The analysis is quite simple. (See below, however.) The previous bound for the first algorithm does seem to have been incorrect.

**Weaknesses:**

The bounds depend on a constant $\kappa$ that can be quite small. The proofs are fairly straightforward.

Proof writing leaves a bit to be desired and I had trouble following some arguments.

1) The constant $\kappa$ and the fact that it is at most $1$ are explained for the first time
2) I believe the Lipschitz constant in Lemma 4 (with the notation employed) should be $L$, or maybe the formula for $T_\beta$ is missing a $1/\epsilon$ factor in the exponent.
3) The last equality in the first math display in page 13 should probably be an upper bound.

**Questions:**

See the above points where I had trouble.

---

> ### Author Response · Authors · 2023-11-22
>
> Thank you for your valuable feedback on our paper. Your insights have been very helpful in refining our work. We have made the following adjustments:
>
> > The constant $\kappa$ and the fact that it is at most $1$ are explained for the first time
>
> A definition $\kappa = 1-\exp(-L\text{diam}\mathcal{X})$ has been added to the sketch proof of Theorem 1.
>
> > I believe the Lipschitz constant in Lemma 4 (with the notation employed) should be $L$, or maybe the formula for $T_{\beta}$ is missing a $1/\epsilon$ factor in the exponent.
>
> This is true, we have missed out an $\epsilon$ in $T_{\mu}$. Please see the updated definition of $T_{\mu}$ before Lemma 4, thanks for pointing it out.
>
>
> > The last equality in the first math display in page 13 should probably be an upper bound.
>
> Sorry about the confusion, it is defined in Lemma 7 that $M(x)=\int_y\exp((f^*(x)+g(y)-C(x,y))/\epsilon)\mathrm{d}\beta(y)$ and this equality is a simplification of the notation.

---

### Official Review · Reviewer_3eZq · 2023-11-02

**Soundness:** 2 fair
**Presentation:** 2 fair
**Contribution:** 3 good
**Rating:** 5
**Confidence:** 2

**Summary:**

This paper deals with computing entropic regularized optimal transport distances between continuous distribution. Traditionally, one draws samples from the continuous distribution and then computes these distances between discrete distribution by constructing an $n^2$ sized cost matrix. However, the focus has shifted on finding the distances between continuous distributions directly. The challenge in doing so is that how one can compactly represent the continuous dual functions in a discrete manner. Mensch and Peyre introduced the online Sinkhorn where they primarily showed how to execute the Sinkhorn algorithm by representing the continuous duals compactly and showed how the dual functions converge to the optimal ones.

There are two main contributions of this paper. First, the authors provide an updated convergence rate for the online Sinkhorn algorithm (after correcting an existing inaccuracy in the work of Mensch and Peyre). They also conduct experiments to suggest that their bound may be tight for certain distributions.

Second, the sample size grows polynomially as the algorithm progresses. To make it more space efficient, they provide a compression mechanism to represent the distributions leading to certain gains in experiments.

**Strengths:**

The problem of estimating dual potentials for OT on continous distribution is a difficult one. For this reason, despite being incremental in nature, I think the result may be important.

**Weaknesses:**

On the negative side, I had a hard time appreciating the five different assumptions made in the paper. I couldn’t quite tell whether they were necessary or they were made as a matter of convenience. Also, the paper is written in a way that makes it only accessible to people who are familiar with previous work (and not for folks who may have a good understanding of the optimal transport problem but lack familiarity with online Sinkhorn). I’m still not able to fully appreciate the result and understand within the landscape of existing algorithms (including time, space and sample complexities etc) for approximating continuous optimal transport. A discussion on this would be good.

Overall, I think this seems like a good contribution, but the writing does not let me full appreciate the result.

**Questions:**

NA

---

> ### Author Response · Authors · 2023-11-22
>
> Thank you for your valuable review of our paper. We've addressed each of your points:
>
> > On the negative side, I had a hard time appreciating the five different assumptions made in the paper. I couldn’t quite tell whether they were necessary or they were made as a matter of convenience.
>
> Assumption 1 on the cost regularity is needed for the OT problem to have a solution.
> The stepsize choices in Assumptions 2 are fairly standard for SGD.
> Assumption 3 provides criteria for the batch size in relation to the accuracy of the Monte Carlo method. We have added further commentary on the assumptions in the text.
>
> The original Assumptions 4 and 5 are made together to ensure that the additional error is limited to $O(t^{b-a})$ and are consistent with the Online Sinkhorn error; please see Section 3.3 for the change we made to these assumptions.
>
> > Also, the paper is written in a way that makes it only accessible to people who are familiar with previous work (and not for folks who may have a good understanding of the optimal transport problem but lack familiarity with online Sinkhorn). I’m still not able to fully appreciate the result and understand within the landscape of existing algorithms (including time, space and sample complexities etc) for approximating continuous optimal transport. A discussion on this would be good.
>
> We have added content in Section 3.1 and discussed the motivation for online Sinkhorn with compression.

---

### Meta-Review · Area_Chair_dhnX · 2023-12-26

**Metareview:**

The authors propose the compressed online Sinkhorn algorithm, an extension of online Sinkhorn (Mensch & Peyré), which goes the extra step (likely needed for this to be truly scalable) of compressing the basis (in addition to proving faster convergence results). The reviewers have all found the contribution relevant, but are all frustrated by the fairly simple nature of experiments, which never really reach practically relevant dimensions for ML. I agree with this assessment, and I believe the paper should be better tailored for a low-$d$ audience (e.g. graphics?), but is not ready in its current form for ICLR. The paper also needs to be further polished (a few mistakes here and there) requiring extra care.

Minor:
- the authors have used a different font to submit their draft. whether intentional or mistaken, I ask them to be careful about this next time, as this is, in itself, a violation of the draft format and sufficient ground for a reject.
- _"Two of the main representations found in the literature include the use of reproducing kernel Hilbert spaces
(Aude et al., 2016) and more recently, the online Sinkhorn algorithm [...]"_ : typo in citation, and I think the authors are missing here all of the many approaches relying on neural networks (e.g. ICNN, Korotin and Makkuva's references.).
- No caption in Figure 2.

**Justification For Why Not Higher Score:**

The paper does not provide convincing experiments (limited to 5D) and still has a long way to go to be publishable.

**Justification For Why Not Lower Score:**

NA

---

### Decision · Program_Chairs · 2024-01-16

Reject